EMBO
Molecular Medicine

# NF45/NF90-mediated rDNA transcription provides a novel target for immunosuppressant development

Hsiang-i Tsai[1,†] (iD), Xiaobin Zeng[2,3,†] (iD), Longshan Liu[4,†], Shengchang Xin[5], Yingyi Wu[1], Zhanxue Xu[1], Huanxi Zhang[4], Gan Liu[1], Zirong Bi[4], Dandan Su[1], Min Yang[1], Yijing Tao[1], Changxi Wang[4], Jing Zhao[5], John E Eriksson[6,7], Wenbin Deng[1,*], Fang Cheng[1,**] & Hongbo Chen[1,***] (iD)

## Abstract

Herein, we demonstrate that NFAT, a key regulator of the immune response, translocates from cytoplasm to nucleolus and interacts with NF45/NF90 complex to collaboratively promote rDNA transcription via triggering the directly binding of NF45/NF90 to the ARRE2-like sequences in rDNA promoter upon T-cell activation *in vitro*. The elevated pre-rRNA level of T cells is also observed in both mouse heart or skin transplantation models and in kidney transplanted patients. Importantly, T-cell activation can be significantly suppressed by inhibiting NF45/NF90-dependent rDNA transcription. Amazingly, CX5461, a rDNA transcription-specific inhibitor, outperformed FK506, the most commonly used immunosuppressant, both in terms of potency and off-target activity (i.e., toxicity), as demonstrated by a series of skin and heart allograft models. Collectively, this reveals NF45/NF90-mediated rDNA transcription as a novel signaling pathway essential for T-cell activation and as a new target for the development of safe and effective immunosuppressants.

**Keywords** CX5461; NF45/NF90; NFAT; nucleolus; organ transplantation
**Subject Category** Immunology

## Introduction

The inhibition of T-cell activation is crucial for both the prevention of organ transplantation rejection and graft-versus-host disease (GVHD) that accompanies allogeneic hematopoietic stem cell transplantation, as well as in the treatment of certain autoimmune diseases (Ichiki *et al*, 2006; Coghill *et al*, 2011; Petrelli & Van Wijk, 2016; Szyska & Na, 2016). The calcineurin-nuclear factor of activated T cells (NFAT) binding inhibitors cyclosporine A (CsA) and tacrolimus (FK506) have proven highly effective at suppressing T-cell response to allografts and are among the most widely used immunosuppressive drugs to significantly prolong graft survival and reduce patient morbidity (Monostory, 2018). The use of these immunosuppressive agents has also been reported in a variety of autoimmune diseases (Kovarik & Burtin, 2003). Despite their widespread application in the clinic, calcineurin inhibitors have been the cause of myriad side effects, including nephrotoxicity, chronic kidney damage, and post-transplant malignancies (Group *et al*, 2018). This may be the result of the general inhibition of calcineurin activity, which plays other biologically important roles besides NFAT activation. The discovery of novel mechanisms of early T-cell activation that obviate the inhibition of calcineurin is therefore of great value in the search for safer and more efficient immunosuppressive agents.

The $Ca^{2+}$-calcineurin-NFAT signaling pathway is a master regulator of T-cell proliferation and activation (Okeefe *et al*, 1992). Five NFAT family members have been identified, namely NFATc1 (also known as NFAT2), NFATc2 (NFAT1), NFATc3 (NFAT4), NFATc4 (NFAT3), and TonEBP (tonicity-responsive enhancer [TonE] binding protein, or NFAT5), among them. NFATC1, NFATc2, and NFATc3 are well-known to play important roles in T-cell activation (Timmerman *et al*, 1997; Macian, 2005; Lee *et al*, 2018). During an adaptive immune response, phosphatase calcineurin dephosphorylates NFAT within T cells. The dephosphorylated NFAT then translocate from the cytoplasm to the nucleus, where they bind directly to antigen response recognition element (ARRE2) within the interleukin-2 (IL-2) enhancer region. This in turn induces IL-2 gene transcription (Jain *et al*, 1993), an essential cytokine for the clonal proliferation and activation of T lymphocytes (Broere & van Eden,

1   School of Pharmaceutical Sciences (Shenzhen), Sun Yat-Sen University, Shenzhen, China
2   Center Lab of Longhua Branch and Department of Infectious Disease, Shenzhen People's Hospital, 2[nd] Clinical Medical College of Jinan University, Shenzhen, China
3   Guangdong Provincial Key Laboratory of Regional Immunity and Diseases, Medicine School of Shenzhen University, Shenzhen, China
4   Organ Transplant Centerm, The First Affiliated Hospital, Sun Yat-sen University, Guangzhou, China
5   State Key Laboratory of Coordination Chemistry, Institute of Chemistry and Biomedical Sciences, School of Life Sciences, Nanjing University, Nanjing, China
6   Cell Biology, Biosciences, Faculty of Science and Engineering, Åbo Akademi University, Turku, Finland
7   Turku Centre for Biotechnology, University of Turku and Åbo Akademi University, Turku, Finland
    *Corresponding author. Tel: +86 15071561390; E-mail: dengwb5@mail.sysu.edu.cn
    **Corresponding author. Tel: +86 18123846151; E-mail: chengf9@mail.sysu.edu.cn
    ***Corresponding author. Tel: +86 15889353410; E-mail: chenhb7@mail.sysu.edu.cn
    † These authors contributed equally to this work

2019). The calcineurin inhibitors CsA and FK506 prevent the calcineurin-driven dephosphorylation of NFAT, thereby inhibiting NFAT nuclear accumulation, IL-2 expression, and the downstream functions of effector T cells.

The nuclear factors NF45 and NF90 were originally isolated in activated Jurkat cells as a heterodimeric complex (NF45/NF90) binding specifically to the ARRE2 enhancer element of the IL-2 promoter (Kao *et al*, 1994). Both NF45 and NF90 have an N-terminal "domain-associated with zinc fingers" (DZF) that resembles template-free nucleotidyltransferases and mediates the heterodimerization of NF45/NF90 through a structurally conserved interface (Wolkowicz & Cook, 2012). Moreover, NF90 has one nuclear localization signal (NLS) domain and two double-stranded RNA binding domains (dsRBDs) in the C-terminal region (Wen *et al*, 2014), which confer binding to highly structured RNAs (Parker *et al*, 2001). In mammals, the NF45/NF90 protein complex is expressed in a wide variety of tissues (Zhao *et al*, 2005) and participates in numerous cellular functions (Shim *et al*, 2002), including cell cycle regulation (Guan *et al*, 2008), transcription activation (Kiesler *et al*, 2010), translational control (Castella *et al*, 2015), DNA damage response (Shamanna *et al*, 2011), microRNA (miRNA) biogenesis (Masuda *et al*, 2013), and viral infection (Idda *et al*, 2019). However, the specific role of NF45/NF90 in the regulation of T-cell activation has not been well established.

In this study, a novel mechanism for the regulation of rDNA transcription in the nucleolus is presented, in which the NF45/NF90 complex plays a key role by interacting with the upstream binding factor (UBF) and promoting the recruitment of RNA Pol I to rDNA promoter regions. We found that the NF45/NF90 regulation of rRNA transcription positively regulates T-cell activation. Suppressing rDNA transcription using the Pol I inhibitor CX5461 significantly inhibited IL-2 secretion, reduced the proliferation of T lymphocytes, and enhanced the survival of mouse skin and heart allografts. This mechanism constitutes a novel therapeutic strategy for immunosuppression.

## Results

### NF45 and NF90 are nucleolar proteins that positively regulate rDNA transcription

Mass spectrometry has been used as a diagnostic tool to confirm the presence of both NF45 and NF90 in the nucleolar proteome (Wandrey *et al*, 2015). In the present study, the nucleolar enrichment of endogenous NF45 and NF90 proteins was confirmed by subcellular fractionation, and an aggregate morphology was observed to colocalize in the nucleolar markers fibrillarin and nucleolin, indicating that NF45 and NF90 are indeed nucleolar proteins compared to nucleoplasm (lamin B1 is a nucleoplasm marker; Fig 1A and Appendix Fig S1A). The primary function of a nucleolus is ribosome biogenesis (including the transcription of rRNAs from rDNAs), the folding, processing, and modification of rRNAs, as well as the assembly of major ribosomal proteins. Intriguingly, the NF45/NF90 heterodimer was recently revealed as a novel regulator of ribosome biogenesis. A luciferase reporter containing the rDNA promoter was therefore used to assess the relationship between NF45/NF90 and rRNA synthesis. Inhibition of

NF45/NF90 by shRNA lentivirus downregulated the luciferase signal, whereas ectopic expression of NF45/NF90 enhanced luciferase activity, suggesting that NF45/NF90 is a positive regulator of rDNA synthesis (Fig 1B and C).

Transcription of the 45S rRNA precursor, 45S pre-rRNA (comprised of the externally transcribed spacers 5′-ETS and 3′-ETS), is a rate-limiting step in ribosome biogenesis, which is followed by the cleavage of 45S pre-rRNA into several smaller rRNAs (18S, 5.8S and 28S rRNAs) (Tiku & Antebi, 2018). We therefore designed primers to target the relevant regions of 45S pre-rRNA in order to measure its rate of transcription (Fig 1D). Consistent with the data from the rDNA promoter-driven luciferase assay, real-time PCR analysis showed that silencing NF45 and NF90 significantly slowed down pre-rRNA transcription, and upregulation of NF45/NF90 dramatically strengthened the process (Fig 1E and F). Silencing NF45/NF90 also suppressed nucleolar FUrd (fluorine-conjugated UTP analogue) incorporation (Appendix Fig S1B), further supporting the regulatory role of NF45/NF90 in rRNA synthesis.

It is known that within the NF45/NF90 heterodimer complex, NF90 contains a DZF motif (NF45 binding domain) and an NLS motif, as well as two RBD domains (RNA binding domains) which can directly bind RNA or DNA (Appendix Fig S2A). In order to investigate the involvement of the individual domains in rRNA synthesis, we inactivated the DZF and RBD domains using various point mutations and truncated the NLS motif to prevent its location in the nucleolus (Appendix Fig S2A–D). Interestingly, the DZF mutant completely reversed the effect of NF90 on transcription of pre-rRNA, indicating that binding to NF45 supports the NF90 regulation of rRNA synthesis (Appendix Fig S2E). Importantly, the NLS mutant also lost the ability to upregulate 45S pre-rRNA levels, demonstrating that the nucleolar localization of NF90 is required for moderating rRNA synthesis (Appendix Fig S2E). Surprisingly, the RBD mutant also resulted in the obvious translocation of NF90 from the nucleolus to the nucleoplasm and downregulated 45S pre-rRNA levels by 12- and 7.7-fold compared to NF90 and the empty vector, respectively (Appendix Fig S2E), suggesting that the RBD mutant of NF90 not only has a lower binding ability to rDNA and rRNA, but also serves as a dominant negative mutant to interfere with wild-type NF45/NF90 functions in the nucleolus.

### NF45/NF90 directly binds rDNA promoters by recognizing ARRE2-like sequence and regulates RNA Pol I transcription machinery

To examine the protein occupancies of NF45 and NF90 at the rDNA loci, chromatin immunoprecipitation (ChIP) experiments were performed (Fig 2A). NF45 and NF90 binding were enhanced by approximately 3- to 5.2-fold and 3.8- to 8.7-fold respectively, compared with the IgG control (Fig 2B), suggesting that NF45/NF90 might regulate rDNA transcription through direct binding to the rDNA loci, especially the rDNA promoter region (H42.9). Since the NF45/NF90 complex has been shown to recognize upstream ARRE2 in the human IL-2 promoter (Shi *et al*, 2007b), the previous report revealed that ARRE-2 site in the IL-2 promoter contains three conserved regions, including a 5'purine NFAT binding site, a AP-1 binding site in the core region, and a 3'purine sequences (Nirula *et al*, 1997). We analyzed the promoter of rDNA gene and found

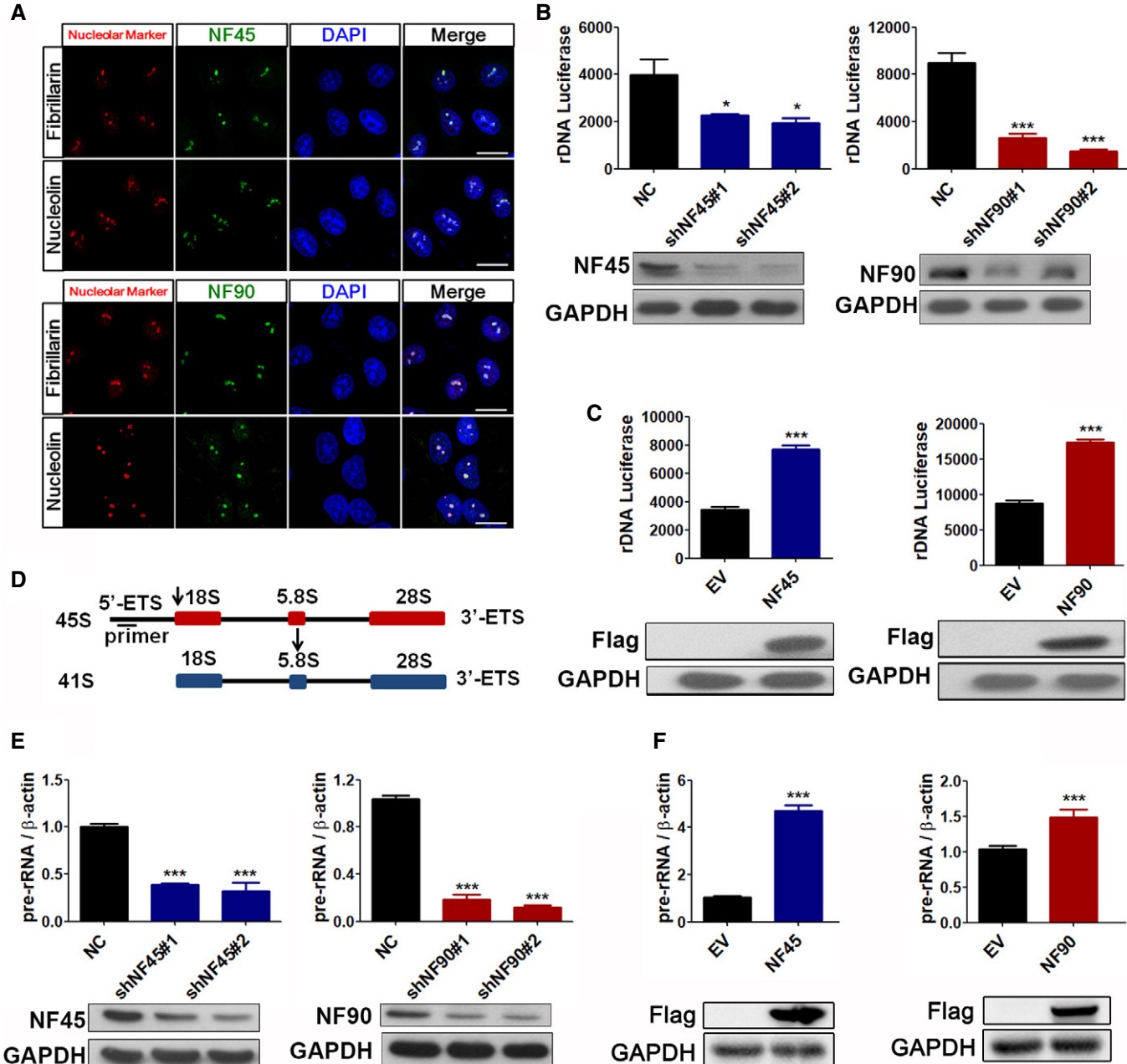

**Figure 1. NF45/NF90 is a nucleolar protein complex that positively regulates rDNA transcription.**

A  Confocal images of HeLa cells stained with NF45/NF90 antibodies and antibodies against nucleolin and fibrillarin. Scale bar, 10 μm.

B  HEK293 cells were infected with lentiviruses containing negative control shRNA (shNC) or NF45/NF90 specific shRNA (shNF45#1, shNF45#2, shNF90#1, and shNF90#2) for 24 h and then transfected with rDNA-Luc. Luciferase activity was measured after a further 24 h (n = 3). Western blots show the protein levels of NF45 and NF90.

C  Empty vector (EV), FLAG-NF45 plasmid (NF45), or FLAG-NF90 plasmid (NF90) were transfected into HEK293 cells for 24 h, and then, rDNA-Luc was cotransfected into the cells. Luciferase activity was measured after a further 24 h (n = 3). Western blots show the protein levels of FLAG-NF45 and FLAG-NF90.

D  The line indicates the position of qPCR primer on the 45S pre-rRNA.

E, F  The qPCR analysis of 45S pre-rRNA from HEK293 cells treated with indicated shRNAs (E) and plasmids (F) for 24 h. Western blots show the expression levels of NF45, NF90, FLAG-NF45, and FLAG-NF90. (n = 3)

Data information: In all panels, bars and error bars represent mean ± SD. Statistical analysis by unpaired Student t-test (C, F). One-way ANOVA analysis followed by Tukey's test (B, E). (*P < 0.05, ***P < 0.001). Exact P values are reported in Appendix Table S3.

Source data are available online for this figure.

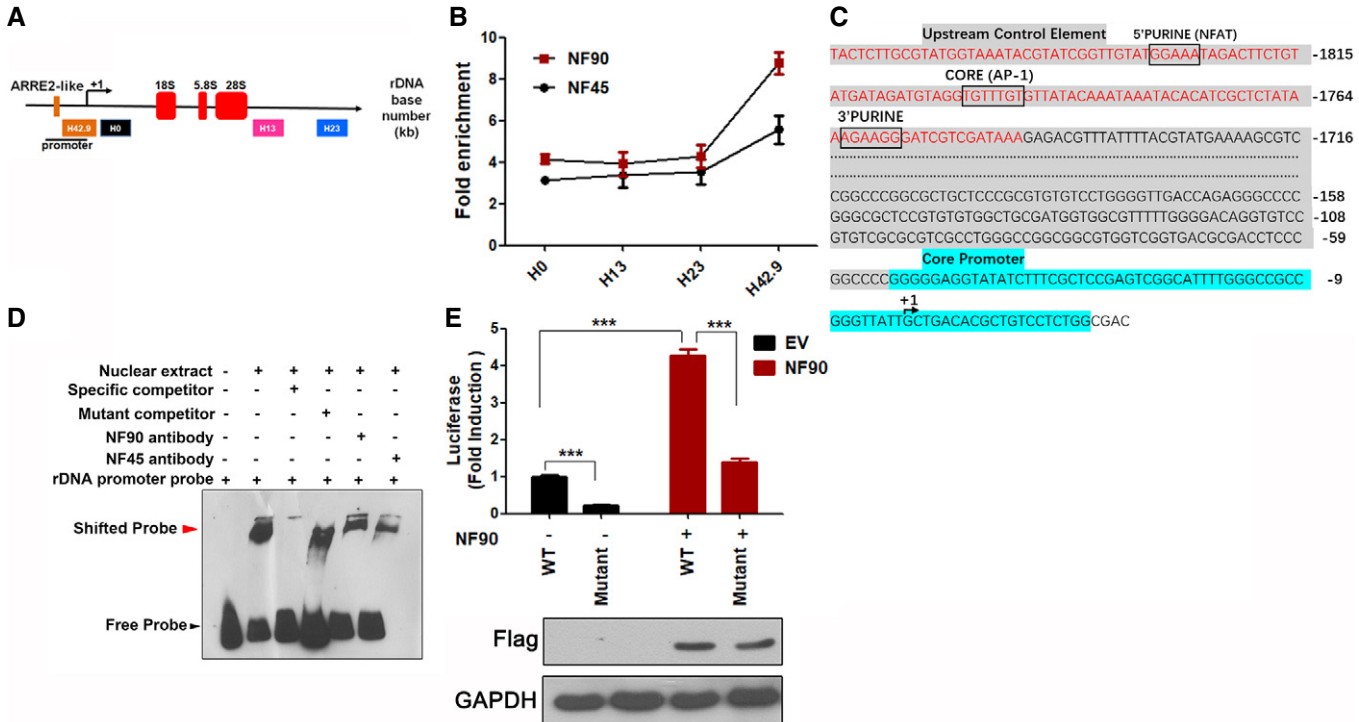

**Figure 2. NF45/NF90 preferentially binds to the promoter region of the rDNA gene loci by directly recognizing the ARRE2-like sequence.**

A  Schematic illustration of a single human rDNA gene repeat and positions of the primers used for ChIP.
B  In Jurkat cells, a ChIP assay was performed with control IgG, NF45, and NF90 antibodies, and then, the precipitated DNA was analyzed using qPCR with the aforementioned primers. The relative rDNA fold enrichment was normalized to control IgG treatment (n = 3).
C  The human rDNA promoter region contains one ARRE2-like sequence. The red letters in gray shaded area indicate the probe sequences used in panel D. Gray shaded area indicates the upstream control element. Blue color indicates the core promoter region.
D  EMSA assay. Lane 1) Biotin-labeled rDNA probe. Lane 2) Biotin-labled rDNA probe + nuclear protein. Lane 3) Biotin-labled rDNA probe + nuclear protein + 100-fold molar excess of biotin-unlabeled specific competitor. Lane 4) Biotin-labled rDNA probe + nuclear protein + 100-fold molar excess of biotin-unlabeled mutant competitor. Lane 5) Biotin-labled rDNA probe + nuclear protein + NF90 antibody. Lane 6) Biotin-labled rDNA probe + nuclear protein + NF45 antibody. The details were described in Materials and Methods section.
E  The relative luciferase activity of HEK293T cells transfected with empty vector (EV) or FLAG-NF90 (NF90) in combination with rDNA-Luc wild-type (WT) or mutated rDNA-Luc reporter plasmid (Mutant) for 24 h. (n = 3) Western blot shows FLAG-NF90 protein level.

Data information: In all panels, bars and error bars represent mean ± SD. Statistical analysis by unpaired Student t-test (E) (***P < 0.001). Exact P values are reported in Appendix Table S3.
Source data are available online for this figure.

that there was a similar ARRE2 region (named ARRE2-like sequence) at nucleotide positions −1,830 to −1,757 from the transcription start site (Fig 2C). As expected, an electrophoretic mobility shift assay (EMSA) and luciferase assay confirmed that the ARRE2-like elements at the rDNA promoter loci are required for NF90 interaction and regulation of rDNA promoter activity (Fig 2D and E).

The regulatory mechanism of rDNA transcription requires the synergistic transaction of UBF followed by promoter selectivity factor 1 (SL1) in order to recruit RNA polymerase I (Pol I) to sit on the rDNA promoter (Fig 3A) (Friedrich et al, 2005). Interestingly, in a co-immunoprecipitation (co-IP) assay, NF45/NF90 had a stronger binding affinity with UBF than a panel of other major nucleolar proteins including S5, L9, B23, and C23 (Fig 3B and C). As UBF activity correlates with phosphorylation and ubiquitination (Zhang et al, 2011), the NF45 and NF90 proteins in HEK293 cells were silenced in order to test their involvement in UBF post-translational modification. In separate experiments, both NF45 and NF90 knockdown decreased the expression level

of total UBF upon serum stimulation (Fig 3D), indicating NF45/NF90 is important for UBF stability. Expectedly, ubiquitination assay found NF90 knockdown caused a K48-linked ubiquitination degradation of UBF protein (Appendix Fig S3A–C). Since UBF activates Pol I to promote rDNA transcription, RNA pol I levels were also examined upon depletion of NF45/NF90. A relative downregulation of RNA pol I expression in shNF45/NF90 cells was observed (Fig 3D), further suggesting that the NF45/NF90 complex positively regulates UBF to recruit RNA pol I to sit on the rDNA promoter.

Previous studies have shown that rDNA transcription can be activated in serum-deprived cells when re-subjected to serum stimulation. To investigate the effects of rRNA transcription on NF45/NF90 localization and function, a ChIP assay was performed. The binding ability of NF45/NF90 to the rDNA promoter was significantly increased following serum stimulation (Fig 3E). CX5461 (Drygin et al, 2011), a highly specific Pol I inhibitor that can suppress rDNA transcription by preventing Pol I-

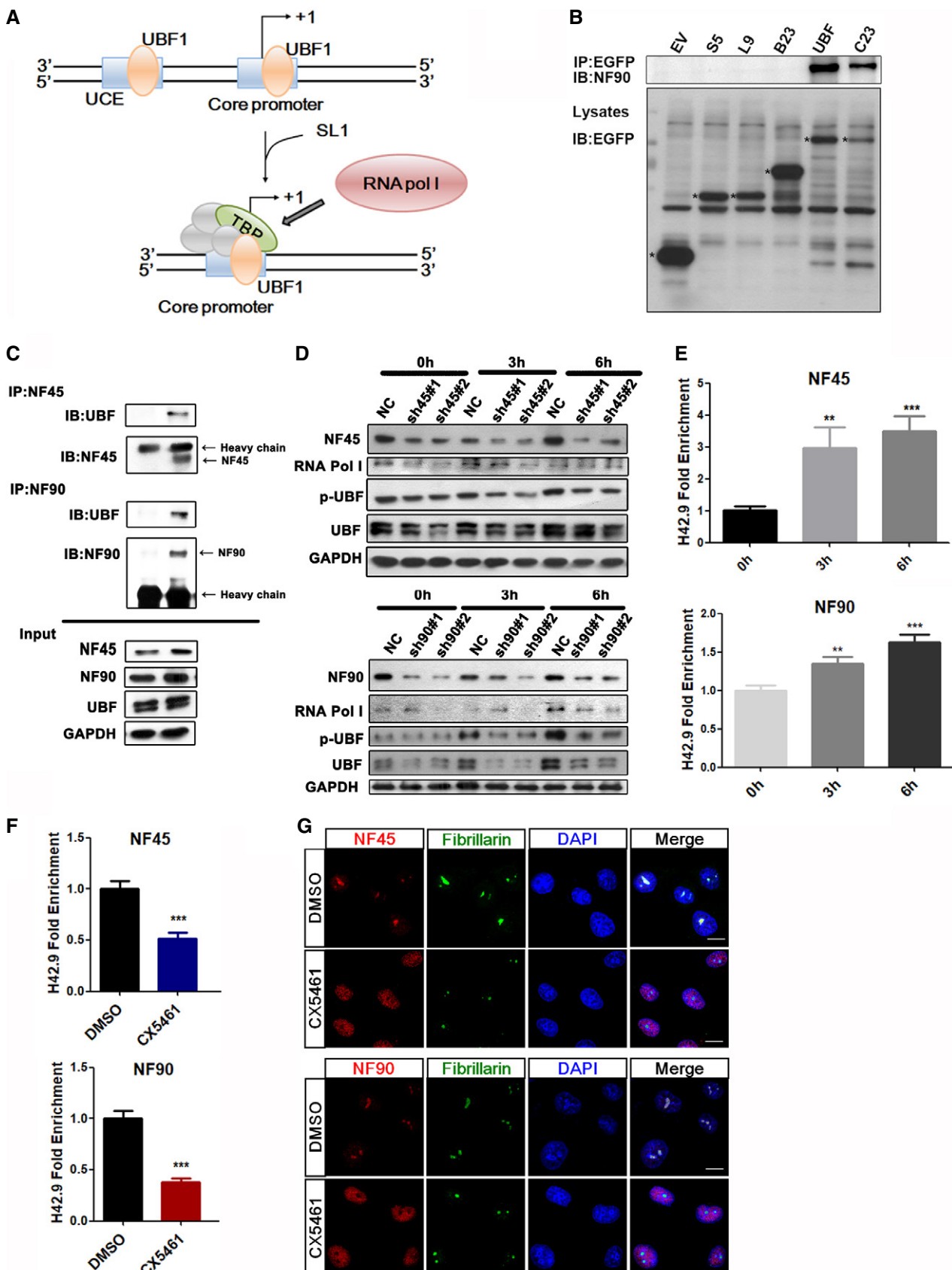

Figure 3.

Figure 3. NF45/NF90 affects rDNA transcription as a positive regulatory factor by recruiting and regulating UBF1 activity.

A The regulatory mechanism of the rDNA transcription. Firstly, UBF1 binds to the upstream control element (UCE) of rDNA promoter to recruit SL1, then SL1 recruits Pol I for rDNA transcription.
B Co-IP analysis of nucleolar proteins with NF90 in HEK293 cells transfected with EV, and the following plasmids: GFP-S5, GFP-L9, GFP-B23, GFP-UBF, and GFP-C23. * represents the target protein.
C Co-IP analysis of endogenous UBF with NF45 and NF90 in HEK293 cells.
D HEK293 cells were infected by lentivirus containing NF45-specific shRNA or NF90-specific shRNA and then stimulated in serum for 0, 3, and 6 h. Cell lysates were prepared to analyze the expression levels of the indicated proteins by Western blot.
E ChIP analysis was used to determinate the binding ability of NF45 and NF90 to rDNA promoters in HeLa cells stimulated with serum for 0, 3 and 6 h. ($n = 3$)
F ChIP analysis of NF45 and NF90 binding to the H42.9 loci in HeLa cells with the treatment of CX5461 or DMSO for 2 h. ($n = 3$)
G Confocal images of HeLa cells treated with CX5461 for 2 h showing NF45 and NF90 colocalized with fibrillarin. Scale bar, 10 μm.

Data information: In all panels, bars and error bars represent mean ± SD. Statistical analysis by unpaired Student $t$-test (F). One-way ANOVA analysis followed by Tukey's test (E) (**$P < 0.01$, ***$P < 0.001$). Exact $P$ values are reported in Appendix Table S3.
Source data are available online for this figure.

specific transcription initiation factors binding to the rDNA promoter to determine the relationship of NF45/NF90 localization and rDNA transcription, was also investigated. In agreement with our hypothesis, the binding ability of NF45/NF90 to the rDNA loci was markedly decreased following treatment with CX5461 (Fig 3F). Interestingly, immunofluorescence analysis revealed a strong translocation of NF45 and NF90 from the nucleolus to the nucleoplasm following CX5461 treatment (Fig 3G), suggesting that the subcellular location of NF45/NF90 is associated with the Pol I-driven transcription machinery.

## NF45/NF90-mediated rDNA transcription contributes to calcineurin-NFAT-mediated T-cell activation *in vitro*

NF45/NF90 was originally isolated in activated Jurkat T cells and is believed to be involved in T-cell activation, but the specific mechanism has yet to be fully elucidated (Kao *et al*, 1994). Here, we postulate that NF45/NF90-mediated rDNA transcription contributes to T-cell activation and proliferation. In agreement with our hypothesis, there was a strong increase of NF45 and NF90 binding to the rDNA promoter when CD3$^+$ T cells were activated with the calcineurin-NFAT stimulators PMA and ionomycin (P/I; Fig 4A). We also noticed that P/I significantly induced 45S pre-rRNA transcription (Fig 4B and C) and rDNA promoter activity detected by luciferase (Fig 4D) and that these were substantially suppressed by NF45/NF90 knockdown (Fig 4C and D).

When T-cell activation was blocked by FK506, NF45 and NF90 translocated from the nucleolus to the nucleocytoplasm and

cytoplasm, respectively (Appendix Fig S4A). This was accompanied by dramatic reductions of NF45/NF90 binding to the rDNA gene loci and 45S pre-rRNA expression (Appendix Fig S4B and C). To elucidate how the calcineurin-NFAT pathway is involved in NF45/NF90-mediated rDNA transcription, the subcellular localization of NFATc1 and NFATc2, two most prominent NFAT family members in T cells, was examined upon T-cell activation (Timmerman *et al*, 1997; Macian, 2005; Lee *et al*, 2018). Both the calcium ionophore ionomycin (Iono) and PMA/Iono (P/I), but not the protein kinase C activator PMA, significantly induced their translocation from cytoplasm to nucleus and nucleolus (Appendix Fig S5A and B). Furthermore, confocal microscopy and co-IP results showed that both NF45 and NF90 colocalized and interacted with NFATc2 and NFATc1 in activated T cells (Fig 4E and F), suggesting that NFAT might interact with the NF45/NF90 complex to cooperatively regulate rDNA transcription in the nucleoli for T-cell activation. In order to confirm the relationship between NF90 and NFAT, we cotransfected NF90 and NFATc2 and found that NF90 and NFATc2 collaboratively promote rDNA transcription after P/I stimulation (Fig 4G). This synergistic effect was inhibited when the ARRE2-like binding site was damaged by deleting GGAAA sequences at the rDNA promoter (Fig 4H). In addition, we found that NFATc2 cannot bind to the rDNA promoter in the absence of NF90 (Fig 4I). Therefore, we speculate that NF90 recruits NFATc2 into the nucleolus to regulate rDNA transcription during T-cell activation.

Next, we questioned whether silencing NF45 or NF90 in T cells would lead to an inhibition of T-cell activation. Fluorescence-activated

Figure 4. NF45/NF90-mediated rDNA transcription contributes to T-cell activation by interacting with NFATc2 in the nucleolus.

A ChIP analysis of NF45 and NF90 binding to the H42.9 loci in CD3$^+$ T cells stimulated with P/I for 6 h ($n = 3$).
B qPCR analysis of pre-rRNA in CD3$^+$ T cells treated with P/I for 6 h ($n = 3$).
C CD3$^+$ T cells were first silenced with shNC, shNF45, or shNF90 and then treated with P/I to extract total RNA for qPCR analysis of pre-rRNA ($n = 3$).
D The analysis of rDNA-luc activity in Jurkat cells with or without NF90 silence upon P/I stimulation or not ($n = 3$).
E Confocal images of Jurkat cells to show NF45 (left panel) or NF90 (right panel) (green fluorescence) colocalized with NFATc2 (red fluorescence) in nucleoli. Scale bar, 10 μm.
F Co-IP analysis of NFATc2 and NFATc1 with NF90 in Jurkat cells treated with P/I.
G ChIP analysis of NFATc2 binding to the rDNA loci in CD3$^+$ T cells stimulated with P/I for 6 h ($n = 3$)
H Jurkat cells cotransfected with rDNA-luc (wild-type or mutant plasmid) with NF90 and/or NFATc2 for 24 h; then, rDNA-luc activity was detected after P/I stimulation or not (DMSO). ($n = 3$)
I ChIP analysis of NFATc2 binding to the H42.9 loci in NF90-silenced CD3$^+$ T cells stimulated with P/I for 6 h. ($n = 3$)

Data information: In all panels, bars and error bars represent mean ± SD. Statistical analysis by unpaired Student $t$-test (A, B, C, D, G, I). One-way ANOVA analysis followed by Tukey's test (H) (*$P < 0.05$, ***$P < 0.001$). Exact $P$ values are reported in Appendix Table S3.
Source data are available online for this figure.

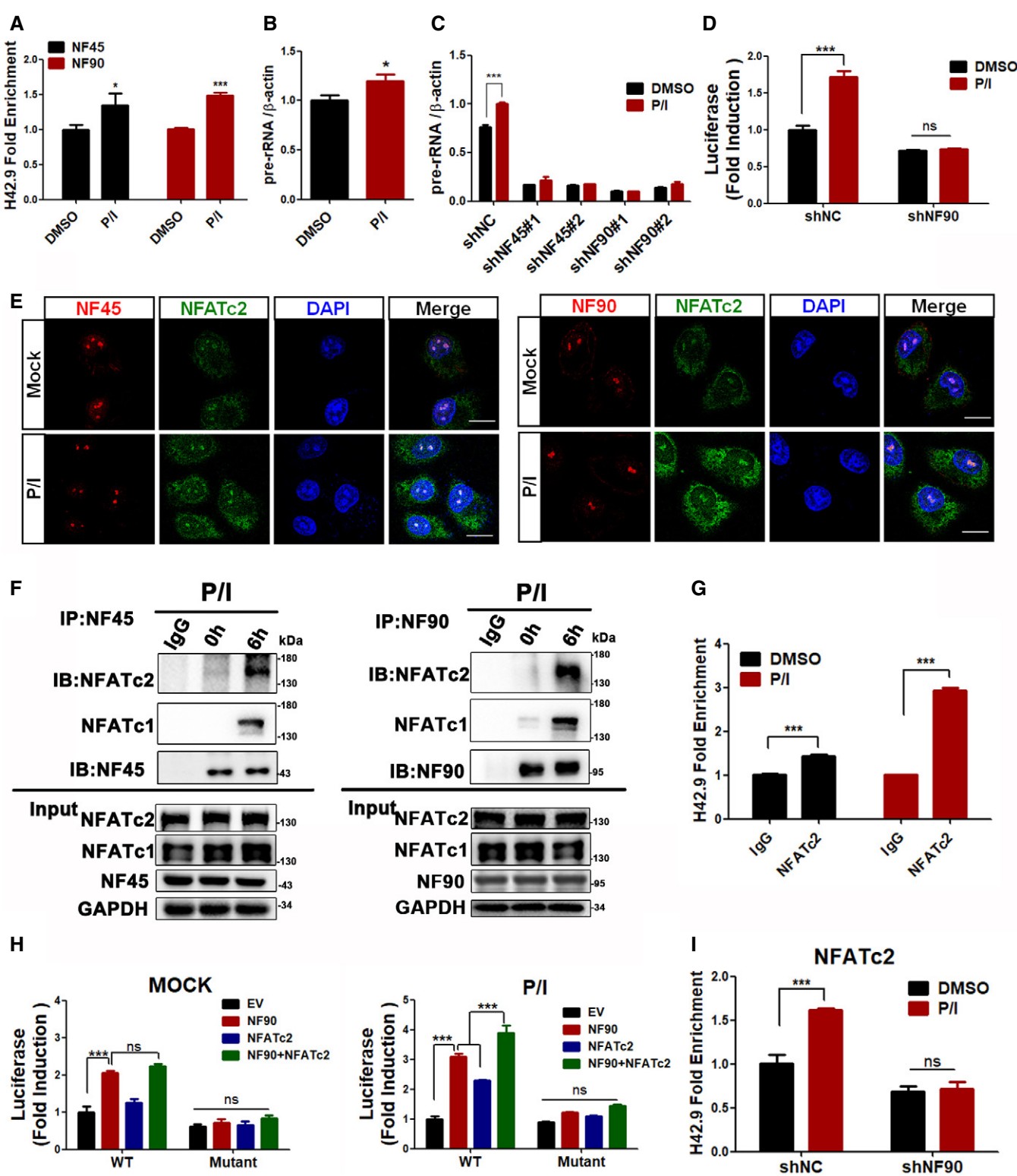

**Figure 4.**

cell sorting analysis (FACS) results suggested that, in the presence of P/I stimulation, NF45/NF90 knockdown significantly inhibited the expression of CD69, an early marker of T-cell activation (Fig 5A). Consistent with this, the secretion protein expression of IL-2 and IFN-γ was also reduced in shNF45 and shNF90 T cells (Fig 5B and

C). Peripheral blood mononuclear cells (PBMC) with NF45 or NF90 silence were also found to have a lower proliferation rate following activation stimulation (Fig 5D). Considered together, these data suggest NF45/NF90-rDNA transcription is necessary for T-cell activation and proliferation.

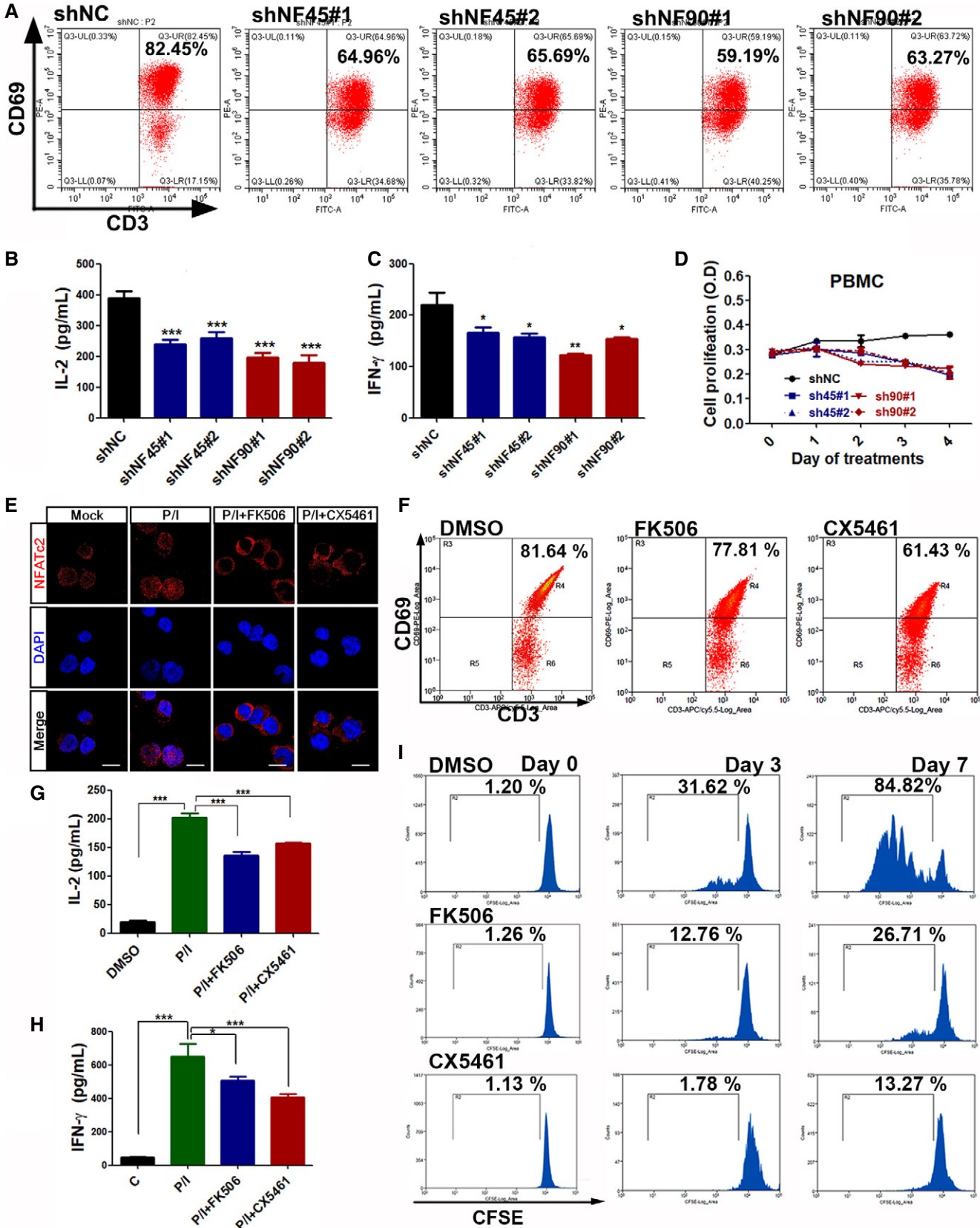

**Figure 5.**

◄

**Figure 5.  NF45/NF90-mediated rDNA transcription is necessary for cell proliferation and activation.**

A       CD69 expression, an early marker of lymphocyte activation, in CD3$^+$ T cells infected with shNC, shNF45, or shNF90 lentivirus upon stimulation with P/I for 12 h.
B, C    PBMC cells with shNC, shNF45, or shNF90 lentivirus infection were stimulated with P/I for 12 h; then, the freshly replaced medium without P/I after another 24 h
        culture was harvested for IL-2 and IFN-γ detections by ELISA (n = 3).
D       The proliferation of shNC, shNF45, or shNF90-infected PBMCs was monitored using a CCK-8 assay (n = 3).
E       Confocal images to show the subcellular localization of NFATc2 in CD3$^+$ T cells treated with P/I and FK506 or CX5461. Scale bar, 10 μm.
F       Flow cytometry analysis of CD69 expression in P/I-stimulated CD3$^+$ T cells treated with or without FK506 or CX5461.
G, H    ELISA assays of IL-2 and IFN-γ in CD3$^+$ T cells with the indicated treatments. (n = 3)
I       Flow cytometry analysis of CFSE-labeled CD3$^+$ T cells in the presence of DMSO, FK506 and CX5461 for 0, 3 and 7 days.

Data information: In all panels, bars and error bars represent mean ± SD. Statistical analysis by One-way ANOVA analysis followed by Tukey's test (E, C, G, H) (*P < 0.05,
**P < 0.01, ***P < 0.001). Exact P values are reported in Appendix Table S3.
Source data are available online for this figure.

Based on the above observations, it is conceivable that T-cell proliferation and activation require cooperation between NFAT-mediated cell factors (such as IL-2 and IFN-γ) and NF45/NF90-mediated ribosomal biogenesis. Thus, inhibiting ribosomal biogenesis might be an alternative pathway to suppress T-cell proliferation and activation. CX5461 is a specific rRNA synthesis inhibitor that selectively inhibits Pol I-driven rRNA transcription and ribosomal biogenesis, but has no effect on Pol II (Drygin *et al*, 2011). Thus, the effects of CX5461 on T-cell activation and proliferation were investigated. Similar to FK506, CX5461 also significantly suppressed the P/I-induced nuclear translocation of NFATc2 from the cytoplasm, indicating a feedback inhibition for NFAT (Fig 5E). Next, T cells were treated with P/I in combination with either FK506 or CX5461. It was found that both CX5461 and FK506 reduced the expression of CD69, IL-2, and IFN-γ (Fig 5F–H), as well as the proliferation of T cells (Fig 5I). Intriguingly, CX5461 had a more dramatic inhibitory effect on T-cell activation and proliferation when compared to FK506, further supporting the key contribution of NFAT-NF45/NF90-mediated ribosomal biogenesis in T-cell activation.

## CX5461 is an effective immunosuppressant and prolongs allograft survival

T-cell activation is one of the key events in acute allograft rejection. It was found that the pre-rRNA level of T cells from recipient mouse spleens was significantly higher in the acute rejection group than in the negative control (NC) group for both skin and heart transplant mouse models (Fig 6A). In order to prosecute this observation in a clinical context, PBMC from renal transplant patients was collected and its pre-rRNA levels were measured. The pre-rRNA levels in T cell-mediated rejection (TCMR) patients were significantly higher than both in antibody-mediated rejection (ABMR) patients and in non-rejection patients after transplantation (Fig 6B). All of these results support the conclusion that pre-rRNA transcriptional activation plays a key role in T-cell activation during allograft rejection.

In order to investigate the feasibility of ribosome biogenesis as a target for the development of immune agents, allogeneic skin graft mice were treated with 2.0 mg/kg FK506, the most commonly used immunosuppressive drug, and different doses of CX5461 (0.5, 1.0, and 2.0 mg/kg, respectively) over a therapeutic window of 30 days (Fig 6C). The body weight of CX5461-treated mice did not change significantly compared to dimethyl sulfoxide (DMSO)-treated control mice, indicating no obvious systemic toxicity for the applied doses (Appendix Fig S7A). However, the FK506 group exhibited obvious weight loss and both sparse and shedding hair around the eyes during treatment, indicating strong off-target activity (i.e., toxicity) in the recipient mice. Obvious skin graft rejection was observed in DMSO-control mice from 5 days after skin transplantation, whereas CX5461 treated mice showed significantly prolonged recipient skin survival in a dose-dependent manner (Fig 6D). Interestingly, treatment with 1.0 mg/kg CX5461 resulted in similar therapeutic kinetics compared to treatment with 2.0 mg/kg FK506, and treatment with 2.0 mg/kg CX5461 had a stronger effect than treatment with FK506 at the same concentration (Fig 6D). Histological examination of graft skins showed massive inflammatory infiltration, obvious edema, and tissue necrosis in the control group. These phenomena

**Figure 6.  CX5461 promotes graft survival in skin and cardiac allograft models.**

▶

A       qPCR analysis of pre-rRNA expression from skin- and heart-transplanted mouse spleens (n = 5).
B       pre-rRNA expression levels of PBMC from renal transplantation patients (Stable group, n = 7. ABMR group, n = 8. TCMR group, n = 8).
C       Therapeutic scheme showing the daily administration of CX5461 or FK506 in skin- or cardiac-allografted mice until day 80.
D       Skin graft survival curve in mice treated with DMSO, 2.0 mg/kg FK506, or 0.5 −2.0 mg/kg CX5461 (n = 5).
E       Representative images of HE staining of skin graft at 7 days post-transplantation. Scale bar, 50 μm.
F       CD3$^+$ T cells (upper) or CD4$^+$ and CD8$^+$ T cells (lower) in the spleens of skin graft recipient mice were analyzed by FACS.
G, H    ELISA analysis of IL-2 and IFN-γ levels in skin graft recipient mouse serum at 4$^{th}$ day post-transplantation (n = 5).
I       The survival time (days) of cardiac graft in mice treated with DMSO, 2.0 mg/kg FK506, or 2.0 mg/kg CX5461 (n = 5).
J       Representative images of HE and CD3-IHC staining of cardiac graft at 7 days post-transplantation. NC, Scale bar, 100 μm.
K       CD3$^+$ T cells (upper) or CD4$^+$ and CD8$^+$ T cells (lower) in the spleens of heart graft recipient mice were analyzed by FACS.
L, M    ELISA analysis of IL-2 and IFN-γ levels in heart graft recipient mouse serum at 7$^{th}$ day post-transplantation (n = 5).
N       qPCR analysis of pre-rRNA expression from transplanted mouse spleen (n = 5).
O       FACS analysis of CD4$^+$, CD25$^+$, and Foxp3$^+$ Treg cells isolated from superficial cervical lymph nodes of heart-transplanted mice.

Data information: In all panels, bars and error bars represent mean ± SD. Statistical analysis by unpaired Student t-test (A). One-way ANOVA analysis followed by
Tukey's test (B, G, H, I, L, M, N) (*P < 0.05, **P < 0.01, ***P < 0.001). Exact P values are reported in Appendix Table S3.
Source data are available online for this figure.

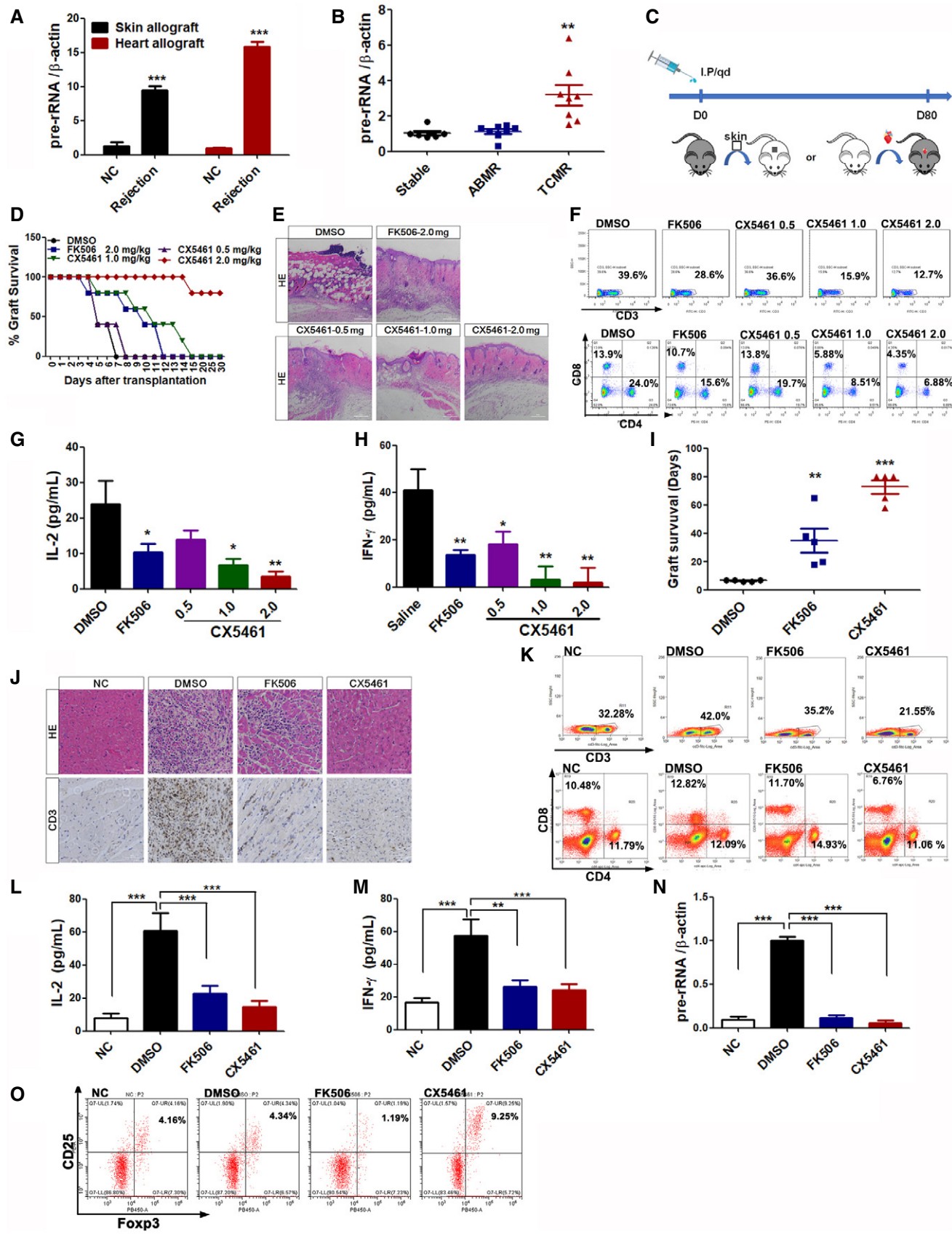

**Figure 6.**

were significantly relieved following CX5461 treatment in a dose-dependent manner, which was also observed in FK506-treated mice (Fig 6E). At the end of the experiment, the mice were sacrificed, and the skin grafts, spleens, and thymuses were separated for further analysis. Significant weight loss was observed only in the thymuses, but not the spleens, in the FK506- and CX5461-treated groups (Appendix Fig S7B), indicating both CX5461 and FK506 cause thymus-specific suppressing activity. Next, mononuclear cells from the spleens were isolated and then labeled them with CD3, CD4, and CD8 monoclonal antibodies for FACS analysis. Treatment with CX5461 significantly decreased the relative percentage of $CD3^+$ T cells (Fig 6F, upper charts) or $CD4^+$ and $CD8^+$ T cells (Fig 6F, lower charts) in the spleens of skin graft recipient mice in a dose-dependent manner. In both cases, 1.0 and 2.0 mg/kg doses of CX5461 were more effective than treatment with FK506 at the same concentrations (Fig 6F).

Commensurate with both these observations and the *in vitro* results, a dose-dependent decrease of both IL-2 and IFN-γ levels in the serum of recipient mice was observed upon treatment with 1.0 and 2.0 mg/kg CX5461. Treatment with FK506 at the same concentrations was less effective (Fig 6G and H).

Recent advances have made the heart transplantation model an excellent tool for quantifying immune rejection by monitoring palpations of the grafted heart (Costello *et al*, 2018). Heart graft mice were further treated with 2.0 mg/kg FK506 and 2.0 mg/kg CX5461, which was the most effective concentration in the skin graft experiments. It was found that CX5461-treated allografts survived significantly longer (> 80 days, animals with live transplants at day 80 were executed, the prescheduled experimental endpoint) than DMSO-treated allografts (6.6 ± 0.5 days) and FK506-treated allografts (41.4 ± 13.7 days; Fig 6I). Similar to skin transplantation experiments, CX5461 treatment had no influence on mouse body weight or other obvious toxic outcomes (Appendix Fig S8A). Similar to skin graft mice, the FK506 group also exhibited weight loss, coarse and pale hairs, and molting around the eyes during treatment, indicating side effects or drug toxicity in the recipient mice (Appendix Fig S8A and B). The hearts, superficial cervical lymph nodes (on the surgical side) and spleens in the DMSO group were significantly larger and heavier than the FK506 and CX5461 groups (Appendix Fig S8C and D). The DMSO-treated group showed severe inflammatory cell infiltration and myocardial necrosis in a histological analysis, whereas the FK506 and CX5461-treated groups showed less lymphocyte infiltration and almost normal tissue structure (Fig 6J). Immunohistochemistry analysis further revealed that CX5461 treatment drastically reduced the infiltration of $CD3^+$ T cells (Fig 6J). Furthermore, treatment with CX5461 decreased the relative percentages of $CD3^+$, $CD4^+$, and $CD8^+$ T cells in the spleens of heart recipient mice, an effect which was more pronounced than treatment with FK506 at the same concentration (Fig 6K). As expected, CX5461 and FK506 treatment reduced the expression of IL-2 and IFN-γ secretion (Fig 6L and M), adding further impetus to the use of CX5461 as an effective immunosuppressive agent. Importantly, pre-rRNA was highly expressed in the spleen lymphocytes of transplanted groups (DMSO) compared to NC mice, whereas nearly complete inhibition was observed when these transplanted mice were treated with CX5461 (Fig 6N). These results suggest that CX5461 suppression of T-cell activation during allograft rejection is associated with its inhibitory effect on rDNA transcription.

As regulatory T cells (Tregs) play an important role in the establishment of immune tolerance in autoimmune diseases and organ transplantation rejection (Alijotas-Reig *et al*, 2014; Hu *et al*, 2016; Romano *et al*, 2017), the $CD4^+$ $CD25^+$ $Foxp3^+$ Treg cell populations isolated from superficial cervical lymph nodes (on the surgical side) were examined. As shown in Fig 6O, consistent with previous reports that FK506 did not increase the proportion of Tregs, even caused inhibition, while CX5461 treatment resulted in a slight but statistically significant increase of Tregs, indicating that CX5461 contributes to immune suppression in the allograft microenvironment partially through regulatory T-cell mechanisms.

## Discussion

In recent years, the NF45/NF90 protein heterodimer has been implicated in a wide variety of biological processes in mammals (Wu *et al*, 2019). NF45/NF90 was previously identified by mass spectrometry to recognize the upstream ARRE2 consensus sequence of the IL-2 promoter in activated T cells, and thus, NF45/NF90 is speculated to mediate T-cell activation by regulating IL-2 transcription (Shi *et al*, 2007a). Herein, we postulate that the main contribution of NF45/NF90 to T-cell activation occurs through the regulation of rDNA transcription, as opposed to IL-2 transcription, for the following reasons. (i) NF45/NF90 is predominantly enriched in the nucleolus, where there is an ARRE2-like site in the rDNA promoter region. (ii) ChIP, EMSA, and luciferase assays confirmed that NF45/NF90 associates with ARRE2-like element of the rDNA promoter (Fig 2B–E). (iii) Importantly, P/I stimulation did not significantly alter the binding affinity of NF45/NF90 to the IL-2 promoter (Appendix Fig S6A), but significantly enhanced the association of NF45/NF90 with the rDNA promoter (Appendix Fig S6A). Moreover, NF90 did not show the same potential as NFAT overexpression to increase the level of IL-2 promoter activity (Appendix Fig S6B).

Ribosome biogenesis is a complex process that requires coordinated transcription and processing of 45S pre-rRNA, the translation, folding, modification, and binding of ribosomal proteins, as well as the assembly and the nucleolar export of the 40S and 60S subunits into the cytoplasm (Strunk *et al*, 2011). A previous study showed that the NF45/NF90 heterodimer binds to the pre-60S ribosome subunit and supports subunit assembly and maturation in the late stages of ribosome biogenesis (Wandrey *et al*, 2015). It is known that the transcription of 45S pre-rRNA is a rate-limiting step in ribosome biogenesis. The assembly of a specific multiprotein complex containing at least UBF, SL1, and transcription initiation factor I (TIF-I) at the rDNA promoter is necessary for the initiation of 45S pre-rRNA synthesis in mammals (Derenzini *et al*, 2017) (Fig 3A). In this study, we found that NF45/NF90 directly binds the rDNA promoter and mediates the interaction of UBF with the rDNA promoter, thus promoting the recruitment of RNA Pol I to the rDNA promoter regions (Fig 7). Overexpressing NF45/NF90 enhanced rDNA promoter activity and 45S pre-rRNA expression, whereas inhibiting NF45/NF90 significantly suppressed these processes (Fig 1B–F). Both nucleolar localization (Fig 1A) and rDNA binding (Fig 2B) of NF90 are required for its role in the transcription of pre-rRNA. Notably, overexpressing NF90 is insufficient to promote pre-rRNA synthesis without binding NF45, suggesting that there is a synergy between NF45 and NF90 in pre-rDNA transcription.

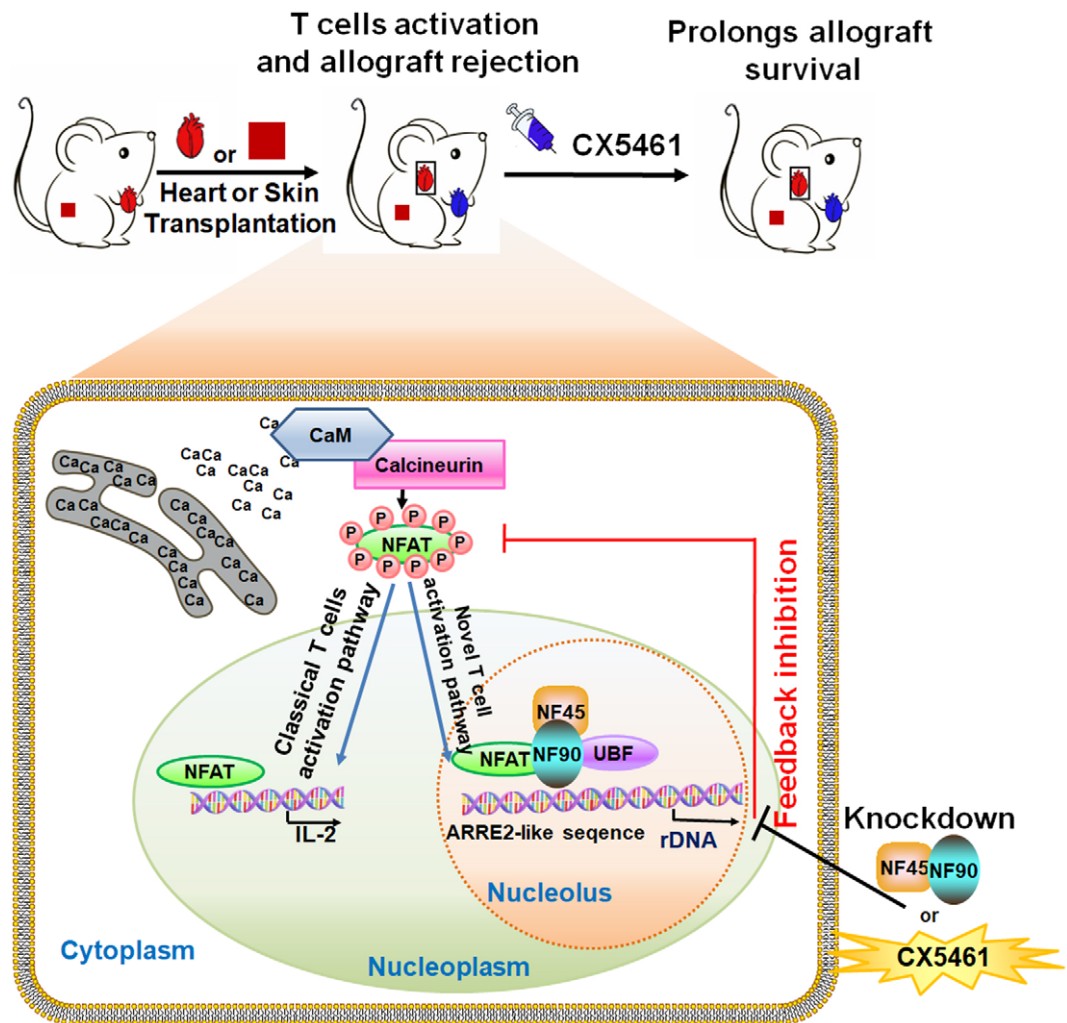

**Figure 7. A schematic model showing the function of NFAT and NF45/NF90 cooperatively promoting rDNA transcription and T-cell activation.**
Besides of the well-known NAFT-IL-2 T-cell activation pathway, NFAT translocates to nucleolus, cooperatively with NF45/NF90 promote rDNA transcription upon T-cell activation. Inhibition of rDNA transcription by NF45/NF90 knockdown or a specific polymerase I inhibitor CX5461 suppressed T-cell activation and skin and heart allograft rejection.

Calcineurin-NFAT-IL-2 axis is a classic signal pathway of T-cell activation. Consistent with previous studies, we also found that P/I stimulation significantly promoted IL-2 promoter activity in T cells overexpressing NFATc1 or NFATc2 (Appendix Fig S6B). However, we also noticed that when T cells were activated, NFAT could not only locate in the nucleus and bind to the IL-2 promoter, but also enter the nucleolus to bind to the rDNA promoter (Appendix Fig S6B), and form a complex with NF45/NF90 (Fig 4E and F) to cooperatively promote rDNA transcription (Fig 4H). We speculate that in resting T cells, NF45/90 can bind to the ARRE2-like site of rDNA and maintain the basic level of rDNA transcription. When T cells are activated, TCR signals could lead to the translocation of NFAT to nucleoli, and NF45/NF90 is the alternative interaction partner for NFAT in nucleoli. The complex formation of NFAT-NF45/NF90-UBF increases rRNA transcription from resting level to activated level. NF45/NF90 knockdown significantly suppressed the binding of NFAT to rDNA promoter (Fig 4I) and T-cell activation and

proliferation (Fig 5A–D), indicating that P/I-induced rDNA transcription is dependent on NF45/NF90 and is necessary for T-cell activation. Thus, we demonstrated that NF45/NF90-mediated rDNA transcription is a novel signaling pathway essential for T-cell activation (Fig 7).

Cyclosporine A and FK506 are two of the most commonly used immunosuppressants in the clinic, but they suffer from side effects, which may be due to intrinsic toxicity or more likely the result of the general inhibition of calcineurin, which plays other biologically important roles besides NFAT activation (Lee *et al*, 2020). It is therefore necessary to develop new immunosuppressive agents based on new targets. Even though ribosome biogenesis occurs in all cells, there is evidence that the selective inhibition of ribosome biogenesis may, in some instances, result in a selective controlling the cell cycle progression to proliferating cells (Derenzini *et al*, 2017). In the process of organ transplantation or autoimmune diseases, T cells are overactivated, which requires the close cooperation of ribosome

biogenesis. As expected, we found that the pre-rRNA levels of T cells in both mouse skin and heart allograft rejection models were significantly higher than in the NC group (Fig 6A). Intriguingly, we also found that the pre-rRNA levels in TCMR patients were also significantly higher than in both ABMR patients and in non-rejection patients after renal transplantation (Fig 6B). Thus, targeting ribosome biogenesis may be a general mechanism to inhibit T-cell overactivation and provide an additional therapeutic strategy for immunosuppression.

CX5461 is an effective and selective anticancer drug, which has been shown to inhibit rRNA transcription by selectively targeting the RNA polymerase I transcription mechanism (Ismael *et al*, 2019). In a phase I clinical trial for advanced hematologic malignancies, CX5461 was safe at doses associated with clinical benefit and dermatologic adverse effects are manageable. Excitingly, antitumor activity was observed in TP53 wild-type and mutant malignancies in this trial, consistent with preclinical data (Clinical trial information: ACTRN12613001061729) (Khot *et al*, 2019). CX5461 is currently in phase I clinical trials for triple negative or BRCA-deficient breast cancer in order to confirm the recommended phase II dose and schedule of CX5461 in patients with solid tumors (Clinical trial identification: NCT02719977) (Hilton *et al*, 2018). Excitingly, we found CX5461 showed that much stronger inhibitory effects than FK506 on T-cell activation and proliferation *in vitro* and *in vivo* and was a more potent prevent of allograft rejection in skin and heart allograft models with lower side effects. For example, CX5461-treated heart allografts survived significantly longer (> 80 days) than DMSO-treated allografts (6.6 ± 0.5 days) and FK506-treated allografts (41.4 ± 13.7 days). In addition, CX5461 treatment also suppressed P/I-triggered NFAT entry into the nucleus, indicating that rDNA transcription inhibition might have a feedback effect on NFAT activation (Figs 5E and 7). This suggests that CX5461 may be a promising immunosuppressant for organ transplant rejection or autoimmune diseases.

In summary, we revealed NF45/NF90-mediated rDNA transcription as a novel signaling pathway essential for T-cell activation and as a new target for the development of safe and effective immunosuppressants.

# Materials and Methods

## Clinical sample of kidney transplant patients

Twenty-five kidney transplant recipients were enrolled in this study. They were divided into three groups: antibody-mediated rejection (ABMR) group ($n = 8$), T cell-mediated rejection (TCMR) group ($n = 8$), and stable group ($n = 7$). The ABMR and TCMR were diagnosed according to the 2015 Banff Kidney Rejection Classification (Loupy *et al*, 2017). Principally, ABMR was diagnosed by the histologic evidence of acute and chronic injury associated with evidence of current/recent antibody interaction with vascular endothelium and serologic evidence of donor-specific antibodies (DSA) to human leukocyte antigen (HLA) or non-HLA antigens, while TCMR was histologically characterized by lymphocytic infiltration of the tubules, interstitium, and, in some cases, the arterial intima. The stable group consisted of kidney transplant recipients in the following conditions: (i) Patients had no proteinuria within the preceding

year or from discharge from the hospital to enrollment in the study, whichever was shorter; (ii) patients' eGFR was > 40 ml/min·1.73 m$^2$ and fluctuated within ± 20% of the mean eGFR within the preceding year or from discharge from the hospital to enrollment in the study, whichever was shorter; and (iii) patients were HLA antibody negative. The demographic and baseline clinical characteristics of the study participants were shown in Appendix Table S1. All the kidney allograft recipients were under "triple regimen" immunosuppression that included the combination of glucocorticosteroids, calcineurin inhibitors (CNIs, FK506, or CsA), and mycophenolic acid (MPA). The dose of CNIs was adjusted based on trough concentration (C0). Approximately 20 ml of heparinized peripheral blood was obtained from the patients at the time of diagnosis and before any kind of treatment (ABMR and TCMR group) or at the follow-up visit (stable group). Written informed consent was obtained from all patients, and the study was approved by the Ethics Committee of the First Affiliated Hospital of Sun Yat-sen University (NO.2019-456) and conformed to World Medical Association Declaration of Helsinki and the Department of Health and Human Services Belmont Report.

## Cell lines, plasmids, antibodies, and chemicals

HEK293, HEK293T, Jurkat, and HeLa cells were purchased from American Type Culture Collection (ATCC). Cells were cultured in RPMI 1640 or DMEM supplemented with 10% fetal calf serum (Hyclone, Logan, UT, USA), 100 units/ml penicillin, and 100 μg/ml streptomycin and were incubated at 37°C in a 5% $CO_2$ atmosphere.

The rDNA promoter-Luc plasmid (rDNA-luc) was constructed by inserting a 2,023 bp sequence (+24 to −1,999 from transcription start site) of rDNA promoter to pHrD-IRES-Luc plasmid, which contained an ARRE2-like site (see Fig 2C). The mutant luciferase construct deleted two NFAT consensus sites of GGAAA at −1,830 to −1826 and −799 to −795. Human NF45 was cloned into the pFLAG-CMV2 vector. Human NF90 and NLS-deleted NF90 were cloned into the pEGFP-c1 vector. Human NFATc1 and NFATc2 were cloned into the pV5-CMV2 vector. The pcDNA3.1-NF90-FLAG, NF90-432/555A (RBD), NF90-T188/T315A (DZF), truncation plasmids were kindly gifted from Professor Honglin Chen (State Key Laboratory for Emerging Infectious Diseases, Department of Microbiology, The University of Hong Kong, Hong Kong). NF45 and NF90 shRNA plasmids were provided by Huang Sidong (McGill University, Canada).

Primary antibodies listed as follows: p-UBF (ser 388): SC-21637, RPA194 (N-17): SC-25931, nucleolin (D-6): sc-17826, UBF (H-300): SC-9131, NF45(G-3): SC-365068, NF90 (N-18): SC-22530, NFATc1 (7A60): SC-7294, UB (P4D1): SC-8017, and Fibrillarin (G-4): SC-66021 were purchased from Santa Cruz with 1:1,000 dilutions. FLAG: M2008M and GAPDH: P30008M were obtained from Abmart (1:5,000, Abmart, Shanghai, China). NFAT1 (D43B1): 5861S and Lamin B1 (D4Q4Z): 12586S were obtained from CST (1:1,000, CST, Boston, USA). HRP- and fluorescein-labeled secondary antibodies were from Gaithersburg (1:5,000). Anti-human CD3 (OKT-3): 10034000, anti-human CD28 (CD28.2): 302902, FITC anti-human CD3 (OKT-3): 317306, PE anti-human CD69 (FN50): 310906, FITC anti-mouse CD3 (17A2): 100204, APC anti-mouse CD4 (GK1.5): 100412, BV510 anti-mouse CD8 (53-6.7): 100752, APC anti-mouse CD25 (PC61): 102012, and Pacific Blue anti-mouse Foxp3 (MF-14): 126410 were from BioLegend (1:100). The ECL detection system was purchased from KPL (Gaithersburg, MD, USA). Anti-FLAG M2

affinity gels (Abmart, Shanghai, China) were used to precipitate FLAG-fused proteins. Dynabeads Protein G magnetic beads used in ChIP assays were purchased from Life Technologies (Carlsbad, CA, USA). CX5461 was purchased from MCE (MedChem express, USA). FK506 was purchased from Selleck Chemicals (Shanghai, China). IL-2 was purchased from Peprotech (Rocky Hill, NJ, USA). Recombinant human interleukin-2 (2 ng/ml, IL-2, Peprotech). Ficoll® lymphocyte separation medium (LSM, MP Biomedicals, Aurora, USA). MojoSortTM Human CD3$^+$ T-cell Isolation Kit, CFSE Division Tracker Kit (5 μg/ml, BioLegend, USA). Cell Counting Kit-8 (CCK-8; Beyotime Institute of Biotechnology).

## Lentiviral shRNA production and infection

For constructs expressing shRNA against NF45 and NF90, in order to produce the shRNA lentivirus, the recombinant packaging plasmids were cotransfected with shNF45 or shNF90 plasmids into HEK293T cells, and culture supernatants containing the virus were collected 48 and 72 h after transfection. For infection with lentivirus, HeLa, PBMC, CD3$^+$ T, and Jurkat cells were cultured with lentivirus solution for 24 h in the presence of 8 μg/ml Polybrene (Sigma).

## Western blotting

Protein fractions were separated by SDS–PAGE and then transferred to PVDF membranes. Membranes were blocked with 5% non-fat milk for 1 h at room temperature and incubated with the desired primary antibodies overnight at 4°C. After 1 h incubation at room temperature with HRP-coupled secondary antibodies, the signal was detected by using an ECL reagent.

## Nucleolus fractionation

Nucleoli were isolated from HEK293 cells as described previously (Hacot et al, 2010). The HEK293 cells were resuspended in 10 volumes of hypotonic buffer (1 mM MgCl$_2$, 10 mM NaCl and 10 mM Tris–HCl, pH 7.4) and incubated on ice for 30 min. Nonidet P-40 (a final concentration of 0.3%) was then added, and the cells were homogenized and lysed using a Dounce homogenizer with a 0.4 mm gap. Nuclei were separated from cytoplasmic fraction by 1,200 g centrifugation for 5 min and then purified by 1,200 g centrifugation for 10 min in a solution containing 880 mM sucros and 5 mM MgCl$_2$. The purified nuclei were resuspended in a solution containing 340 mM sucrose and 5 mM MgCl$_2$ and sonicated several times. Each sonication lasted for 30 s, with an interval of 5 min, until the cell nuclear membrane completely ruptured, but the nucleolus was intact under microscopy. The nucleoli and nucleoplasm (supernatant) were collected by 2,000 g centrifugation in 880 mM sucrose for 20 min. The nuclear, nucleoplasmic, and nucleolar fractions were subjected to Western blotting using the antibodies indicated.

## Quantitative PCR

Total RNA was extracted using the TRIzol regent (Invitrogen Corp., Carlsbad, CA, USA) according to the manufacturer's specification. The cDNA was synthesized by reverse transcription using random primers, and the product was used to analyze mRNA using SYBR Green real-time quantitative PCR (qPCR; Biotool, China). qPCR primers are listed in Appendix Table S2.

## Luciferase reporter assay

Cells were cotransfected with wild-type or mutant rDNA promoter-Luc, shNF90, pcDNA3.1-NF90-FLAG, or pCMV3-ORF-V5 for 48h. Cells were then harvested in the lysis buffer, and the relative rDNA promoter activity was measured in cell lysates using a dual luciferase assay kit (https://worldwide.promega.com/products/lucifera se-assays/reporter-assays/dual_luciferase-reporter-assay-system/?ca tNum=E1910, Promega, Madison, WI, USA).

## Immunofluorescence assay

Cells growing on glass coverslips were fixed with ice methanol for 10 min and then blocked with 3% BSA containing 0.4% Triton X-100 for 1h. After overnight incubation at 4°C with primary antibodies, the coverslips were incubated with fluorescein or rhodamine-conjugated secondary antibodies for 1 h at room temperature. Nuclei were counterstained with DAPI and imaged using a confocal microscope.

## 5-Fluorouridine (Furd) incorporation assay

Silenced of NF45, NF90, or control HeLa cells grown on coverslips in a 12-well culture plate. FUrd (Sigma-Aldrich, St. Louis, MO, USA) was added a final concentration of 10 mM for 15 min, cells were subjected to immunodetection with BrdU primary antibody at 4°C overnight, and the coverslips were incubated with rhodamine-conjugated secondary antibodies for 1h at room temperature. Nuclei were counterstained with DAPI and imaged using a confocal microscope.

## Co-immunoprecipitation assay

Cells were lysed in an RIPA buffer (150 mM NaCl, 0.1 % Triton X-100, 0.5% sodium deoxycholate, 0.1% SDS, 50 mM Tris–HCl, pH 8.0, nuclease, protease inhibitor cocktail) and subjected to sonication for 10 s. The cell lysates were clarified using centrifugation at 12,000 g for 15 min. Protein extracts were then mixed with the indicated primary antibodies and protein A/G agarose (Sigma, St. Louis, MO, USA) or anti-FLAG M2 agarose for overnight at 4°C. The complexes were collected and washed three times with TBST (1X Tris-Buffered Saline, 0.1% Tween® 20 Detergent). The resolved proteins were analyzed using Western blotting.

## Chromatin immunoprecipitation

Chromatin immunoprecipitation was performed in an accordance with a published protocol (Nelson et al, 2006) with minor modifications. Briefly, cells were fixed with 1% formaldehyde for 15 min at room temperature. Cells were collected and resuspended in a ChIP lysis buffer (50 mM Tris–HCl pH 7.5, 150 mM NaCl, 5 mM ethylenediaminetetraacetic acid [EDTA], 0.5% NP-40, 1.0% Triton X-100, and protease inhibitor cocktail) and then subjected to sonication to produce 200–1,000 bp DNA fragments. Sheared chromatin was immunoprecipitated with the indicated antibodies or control IgG in combination with Dynabeads Protein G magnetic beads. After several washes, the beads were resuspended with 10% Chelex 100 suspension and boiled for

10 min. Then, samples were incubated using Proteinase K was added the next day and incubated in a shaker (1 h, 55°C at 1,400 $g$) followed by boiling for 10 min. After centrifugation, the suspension was collected to perform qPCR analysis, and fold enrichment relative to the IgG control antibody was calculated. Primers (H0, H13, H23, H42.9) for the rDNA loci used in the ChIP assays were synthesized as specified previously (Grandori et al, 2005).

### Electrophoretic mobility shift assays

Nuclear extracts of HEK293 cells transfected with pcDNA3.1-FLAG-NF90 were prepared according to the protocol of Preparing Nuclear and Cytoplasmic Extracts (Lahiri & Ge, 2000). Non-radioactive EMSA was performed by using EMSA/Gel-Shift Kit (Beyotime Biotechnology, Shanghai, and China). Probe for EMSA was obtained from the rDNA promoter-luc construct by PCR using the following biotin-labeled primers.

Forward: 5′-TGGAGACACGGGCCGGCCCCCT-3′;
Reverse: 5′-TTTATCGACGATCCCTTCTTTA-3.

The sequence of probe matches the red letters on gray shaded area in Fig 2C. Specific competitor and mutant competitor were also obtained from the rDNA-luc or the mutant rDNA-luc construct by PCR using the above biotin-unlabeled primers. Nuclear extracts were incubated with the following mixture samples: (i) biotin-labeled rDNA probe, (ii) biotin-labeled rDNA probe + 100-fold molar excess of biotin-unlabeled specific or mutant competitor, and (iii) biotin-labeled rDNA probe + NF90 or NF45 antibody. Where competition reactions are required, incubate with the unlabeled competitor for 15 min or NF45/NF90 antibodies for 1 h at room temperature before adding the biotin-labeled probe. The reactions were transferred into a pre-run 6% native polyacrylamide mini-gel and submitted to gel electrophoresis. Following electrophoretic separation, oligonucleotide–protein complexes were transferred by electro-blotting (330 mA, 1 h) onto specific positively charged nylon membranes (Bio-Rad) and detected using an anti-digoxigenin antibody conjugated with alkaline phosphatase.

### Isolation of PBMC and CD3$^+$ T cells

Peripheral blood from normal, healthy donors was collected into potassium EDTA vacates. Peripheral blood mononuclear cells (PBMCs) were isolated using Ficoll® lymphocyte separation medium (LSM, MP Biomedicals, Aurora, USA) centrifugation as described previously (Chikamatsu et al, 2007). Next, CD3 T cells were separated from freshly PBMCs by using a MojoSort™ Human CD3$^+$ T cell Isolation Kit (BioLegend, https://www.biolegend.com/protocols/mojosort-isolation-kits-protocol-1/4599/).

For CD69 early activation detection, freshly CD3$^+$ T cells were infected with shNC, shNF45 or shNF90 lentivirus; then, cells were cultured in plates coated with anti-CD3 and -CD28 mAbs and stimulated with P/I for 12 h. The CD3-positive T cells were further pre-gated with anti-CD3 antibody, and CD69 expression was analyzed using an anti-CD69 antibody by FACS.

### CCK-8 cell proliferation assay

PBMCs proliferation was evaluated using the Cell Counting Kit-8 (CCK-8; Beyotime Institute of Biotechnology) and following the manufacturer's instructions. Briefly, ShNC, shNF45, or shNF90 lentivirus-infected PMMCs were seeded on 96-well plates coated with anti-CD3 and -CD28 mAbs and cultured in R10 medium (RPMI supplemented with 10% FBS, L-glutamine, penicillin/streptomycin, and 10 ng/ml IL-2). Then, 10 μl CCK-8 solutions were added to each well at different time points and incubated for 2 h. The absorbance at 450 nm was measured using a microplate reader. Results are representative of three individual experiments in triplicate.

### CFSE cell proliferation assay

Freshly isolated CD3$^+$ T cells were labeled using CFSE working solution (5 μM) using the CFSE Division Tracker Kit (BioLegend, USA). T cells were incubated for 20 min at 37°C and quenched staining with culture medium. Next, cells were seeded in 24-well plate coated with anti-CD3 and -CD28 mAbs. At days 0, 3 and 7, T cells were collected and subjected to FACS (Beckman Coulter, Brea, CA, USA) using the Cell Quest software (BD Biosciences, Mountain View, CA, USA).

### Skin allograft and heart transplantation models

Male BALB/c mice were obtained from the Institute of Laboratory Animal Sciences, Chinese Academy of Medical Science. All animal studies were carried out according to the protocols approved by the Administrative Committee on Animal Research of Sun Yat-sen University. The approved numbers were 2019000349 and 2019000352. All mice were maintained in a pathogen-free environment at 21 ± 2°C and 20% humidity with a 12-h light/dark cycle. The animals were acclimatized for one week prior to use. All experiments were performed under laminar flow hoods. C56BL/6 skin transplantation was performed in eight-week-old male BALB/c mice according to a standard procedure. Skin grafts (1 cm$^2$ in size) from C56BL/6 mice were dorsally transplanted onto recipient BALB/c mice. All 48 recipients were randomly divided into four equal groups: control group (DMSO, $n = 10$), group 2 (CX5461 0.5 mg/kg/day, $n = 10$), group 3 (CX5461 1.0 mg/kg/day, $n = 10$), and group 4 (CX5461 2.0 mg/kg/day, $n = 10$), group 5 (FK506 2.0 mg/kg/day, $n = 10$). Three days prior to skin transplantation, the recipients were treated with medicines via intraperitoneal injection every day until 12 days after transplantation. The allografts were observed and photographed daily for 30 days, in which the necrosis mice were calculated, where > 90% necrosis of the skin allograft was defined as rejection. Graft necrosis was grossly determined based on graft appearance, color, texture, and absence of bleeding when cut with a scalpel. The mean survival time (MST) of skin grafts in all groups was calculated.

To carry out heart transplantation, hearts from male BALB/c mice as donors were transplanted into the cervices of male C57BL/6 mice (8–10 weeks old, weight > 22 g). The doses and injection time of all 30 recipients were randomly divided into four equal groups: control group (DMSO, $n = 10$), group 2 (CX5461 2.0 mg/kg/day, $n = 10$), and group 3 (FK506 2.0 mg/kg/day, $n = 10$). Three days prior to heart transplantation, the recipients were treated with medicines via intraperitoneal injection every day until 80 days after transplantation. Transplanted mice were sacrificed at 7 days to dissect graft-heart, superficial cervical lymph nodes (on the surgical side) and spleen followed by downstream analysis.

## The paper explained

### Problem

T cells are highly activated and play a central role for transplant rejection. Two potent immunosuppressants, cyclosporin A, and FK506, block the calcineurin-NFAT-dependent T-cell activation in organ transplantation. However, the serious side effects limit the clinical use of both drugs. Clinically applicable strategies for inhibiting T-cell activation through alternative regulatory mechanisms need to be developed to achieve long-term graft tolerance. The heterodimeric nuclear complex NF45/NF90 binds to an NFAT response element in the IL-2 promoter and regulates IL-2 activation in Jurkat T cells. The current study explores whether and how NF45/NF90 complex mediate T-cell activation in experimental transplantation.

### Results

Here, we show that the NF45/NF90 complex is preferentially localized in the nucleoli and directly binds to an ARRE2-like sequence in rDNA promoter. Upon T-cell activation, NFAT translocates from cytoplasm to the nucleolus and cooperatively promotes rDNA transcription by interacting with NF45/90. NF45/NF90 knockdown significantly suppresses rDNA transcription and T-cell activation, indicating NF45/NF90-mediated rDNA transcription plays a key role in the classical calcineurin-NFAT T-cell activation pathway. Excitingly, a specific rRNA synthesis inhibitor CX5461 shows a more dramatic inhibition effect on T-cell activation and leads to a longer survival time of skin and heart allografts than the most commonly used calcineurin inhibitor FK506.

### Impact

Our study offers insights into the role and molecular mechanism of NFAT and NF45/NF90 in modulating T-cell activation. We propose that preventing rDNA transcription and ribosome biogenesis may be a promising strategy to prevent organ transplant rejection.

### ELISA detection of cytokines

The mice serum or cell culture medium were collected, and the concentrations of IL-2 and IFN-γ were determined using an ELISA kit (Dakewe Biotech Inc., Co. Ltd., Shenzhen, China), by using recombinant cytokines to establish standard curves.

### Histological and immunohistochemistry analysis

Seven days after surgery, the mouse recipients were sacrificed and tissue specimens from the transplantation site were collected, fixed with 4% paraformaldehyde for 24 h, and embedded with paraffin. Sections (4 μm thickness) were stained with hematoxylin–eosin for conventional morphological evaluation. Ten visual fields of each sample were randomly selected to observe the inflammatory cell infiltration. For the immunohistochemical staining, deparaffinized tissue sections were incubated with mouse monoclonal CD3 (1:100, PC3/188A: sc-20047, Santa Cruz Biotechnology), followed by visualization using HRP/DAB detection IHC Kit and counterstaining with Mayer's hematoxylin.

### Flow cytometric analysis for T cells in spleens and lymph nodes

For the percentage analysis of CD3[+], CD4[+], and CD8[+] T cells, the recipient mouse spleens were harvested under sterile conditions and gently sheared in a sterile cell dish. A single-cell suspension was prepared by passing grinding fluid through a 70-mesh cell sieve, and the erythrocytes were lysed using a red blood cell lysis buffer and then were labeled with FITC-CD3, APC-conjugated CD4 and BV510-conjugated CD8 monoclonal antibodies. Lymphocytes were first gated by typical FSC and SSC characteristics and subsequently subjected to CD3, CD4, and CD8 analysis by FACS.

For the percentage analysis of CD4[+] CD25[+] Foxp3[+] Tregs, the superficial cervical lymph nodes on the surgical side of recipient mice were harvested and a single-cell suspension was prepared as described above. Cells first treated with APC-labeled anti-mouse CD4 and PE-labeled anti-mouse CD25 antibodies, then fixed by fixing buffer for 20 min, permeated with True-Nuclear™ Transcription Factor Buffer (BioLegend), and then stained with Pacific Blue anti-mouse Foxp3 antibody. Lymphocytes were first gated by typical FSC and SSC characteristics and subsequently subjected to CD4 and CD25 and Foxp3 analysis by FACS.

### Statistical analysis

Data were expressed as means ± standard deviation (SD) of at least three independent experiments. The results were analyzed by one-way analysis of variance (ANOVA) followed by Turkey test and statistical analysis by unpaired Student t-test. All statistical procedures were performed with GraphPad Prism ver 5.0 software (GraphPad Software). The exact p-values are listed in Appendix Table S3.

## Data availability

Human clinical data are available via e-mail request. This study includes no data deposited in external repositories.

**Expanded View** for this article is available online.

## Acknowledgements

This research was supported by the National Natural Science Foundation of China (81702750, 81670141, 81970145, 81700655 and 82001698); Natural Science Foundation of Guangdong Province (2020A1515011465 and 2020A151501467); Science, Technology & Innovation Commission of Shenzhen Municipality (JCYJ20170818164756460, JCYJ20180307154700308, JCYJ20170818163844015, JCYJ20180307151420045, and JCYJ20190807151609464); Sun Yat-sen University (20ykzd17 and 20ykpy122); International Collaboration of Science and Technology of Guangdong Province (2020A0505100031); China Postdoctoral Science Foundation (2018M643299).

## Author contributions

HC, FC, WD, and H-iT designed the experiments and wrote the manuscript. H-iT, XZ, LL, SX, YT, and YW performed in vitro experiment. H-iT, ZX, HZ, GL, ZB, DS, and MY performed in vivo experiments. CW, JZ, and JEE made a lot of suggestions on the experimental scheme and made contributions to the analysis of in vivo and in vitro data. HC, FC, and HT wrote the paper.

## Conflict of interest

The authors declare that they have no conflict of interest.

## For more information

The website of Dr. Hongbo Chen and Fang Cheng: https://www.x-mol.com/groups/chenhb_chengf

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
