## [Review Process File · EMBO Molecular Medicine]

NF45/NF90-mediated rDNA transcription provides a novel target for immunosuppressant development

Tsai Hsiang-i, Xiaobin Zeng, Longshan Liu, Shengchang Xin, Yingyi Wu, Zhanxue Xu, Huanxi Zhang, Gan Liu, Zirong Bi, Dandan Su, Min Yang, Yijing Tao, Changxi Wang, Jing Zhao, John Eriksson, Wenbin Deng, Fang Cheng, and Hongbo Chen

DOI: [10.15252/emmm.202012834](https://doi.org/10.15252/emmm.202012834)

Corresponding author(s): [Hongbo Chen \(chenhb7@mail.sysu.edu.cn\)](mailto:chenhb7@mail.sysu.edu.cn)

Review Timeline:

Submission Date:	29th May 20
Editorial Decision:	30th Jun 20
Revision Received:	30th Nov 20
Editorial Decision:	11th Dec 20
Revision Received:	26th Dec 20
Accepted:	4th Jan 21

Editor: *Jingyi Hou*

Transaction Report:

30th Jun 2020

Dear Dr. Chen,

Thank you for the submission of your manuscript to EMBO Molecular Medicine. We have now received feedback from the two referees whom we asked to evaluate your manuscript. As you will see from the reports below, the referees acknowledge the potential interest of the study. However, they also raise substantial concerns about your work, which should be convincingly addressed in a major revision of the present manuscript.

I think that the referees' recommendations are rather clear and there is no need to reiterate their comments. The clarity in data/study presentation needs to be improved, more details and information need to be provided, additional experiments and controls should be performed to strengthen the molecular mechanism as suggested by referee #2.

We would welcome the submission of a revised version within three months for further consideration. Please note that EMBO Molecular Medicine strongly supports a single round of revision and that, as acceptance or rejection of the manuscript will depend on another round of review, your responses should be as complete as possible.

We are aware that many laboratories cannot function at full efficiency during the current COVID-19/SARS-CoV-2 pandemic and have therefore extended our "scooping protection policy" to cover the period required for a full revision to address the experimental issues. Please let me know should you need additional time, and also if you see a paper with related content published elsewhere.

I look forward to receiving your revised manuscript.

Yours sincerely,
Jingyi Hou

Jingyi Hou
Editor
EMBO Molecular Medicine

*** Instructions to submit your revised manuscript ***

**** PLEASE NOTE **** As part of the EMBO Publications transparent editorial process initiative (see our Editorial at <https://www.embopress.org/doi/pdf/10.1002/emmm.201000094>), EMBO Molecular Medicine will publish online a Review Process File to accompany accepted manuscripts.

To submit your manuscript, please follow this link:

Link Not Available

- 1) a .docx formatted version of the manuscript text (including Figure legends and tables). Please make sure that the changes are highlighted to be clearly visible to referees and editors alike.
- 2) separate figure files*
- 3) supplemental information as Expanded View and/or Appendix. Please carefully check the authors guidelines for formatting Expanded view and Appendix figures and tables at <https://www.embopress.org/page/journal/17574684/authorguide#expandedview>
- 4) a letter INCLUDING the reviewers' reports and your detailed responses to their comments (as Word file)

Also, and to save some time should your paper be accepted, please read below for additional information regarding some features of our research articles:

- 5) The paper explained: EMBO Molecular Medicine articles are accompanied by a summary of the articles to emphasize the major findings in the paper and their medical implications for the non-specialist reader. Please provide a draft summary of your article highlighting
 - the medical issue you are addressing,
 - the results obtained and
 - their clinical impact.

- 6) For more information: There is space at the end of each article to list relevant web links for further consultation by our readers. Could you identify some relevant ones and provide such information as well? Some examples are patient associations, relevant databases,

OMIM/proteins/genes links, author's websites, etc...

7) Author contributions: the contribution of every author must be detailed in a separate section (before the acknowledgments).

8) EMBO Molecular Medicine now requires a complete author checklist (<https://www.embopress.org/page/journal/17574684/authorguide>) to be submitted with all revised manuscripts. Please use the checklist as a guideline for the sort of information we need WITHIN the manuscript as well as in the checklist. This is particularly important for animal reporting, antibody dilutions (missing) and exact p-values and n that should be indicated instead of a range.

9) Every published paper now includes a 'Synopsis' to further enhance discoverability. Synopses are displayed on the journal webpage and are freely accessible to all readers. They include a short stand first (maximum of 300 characters, including space) as well as 2-5 one sentence bullet points that summarise the paper. Please write the bullet points to summarise the key NEW findings. They should be designed to be complementary to the abstract - i.e. not repeat the same text. We encourage inclusion of key acronyms and quantitative information (maximum of 30 words / bullet point). Please use the passive voice. Please attach these in a separate file or send them by email, we will incorporate them accordingly.

You are also welcome to suggest a striking image or visual abstract to illustrate your article. If you do please provide a jpeg file 550 px-wide x 400-px high.

10) A Conflict of Interest statement should be provided in the main text

11) Please note that we now mandate that all corresponding authors list an ORCID digital identifier. This takes <90 seconds to complete. We encourage all authors to supply an ORCID identifier, which will be linked to their name for unambiguous name identification.

Currently, our records indicate that the ORCID for your account is 0000-0002-0954-5600.

Link Not Available

12) The system will prompt you to fill in your funding and payment information. This will allow Wiley to send you a quote for the article processing charge (APC) in case of acceptance. This quote takes into account any reduction or fee waivers that you may be eligible for. Authors do not need to pay any fees before their manuscript is accepted and transferred to our publisher.

Photos 400-800 DPI

Figures are not edited by the production team. All lettering should be the same size and style; figure

panels should be indicated by capital letters (A, B, C etc). Gridlines are not allowed except for log plots. Figures should be numbered in the order of their appearance in the text with Arabic numerals. Each Figure must have a separate legend and a caption is needed for each panel.

*Additional important information regarding figures and illustrations can be found at <http://bit.ly/EMBOPressFigurePreparationGuideline>

***** Reviewer's comments *****

Referee #1 (Remarks for Author):

In this manuscript the authors presented a new pathway for the NF45/NF90 complex, which was initially known to mediate rDNA transcription. They demonstrated with very convincing data that NF45/NF90 activation contributes to T cell activation and could be blocked by the transcription inhibitor CX5461. They showed successful in vitro and in vivo blockade of T cell activation by CX5461 which may prevent graft-versus-host disease.

The manuscript is well written and the experimental setup is comprehensible. The authors added further knowledge to the role of the NF45/NF90 complex, which is important to the field as well as for less specialized audience. I am not sure if the authors are aware of the clinical trials with CX5461, which would further strengthen their postulated use in a clinical setting.

Minor points:

- Data analyzing T cell population (Figure 6) could be supported by absolute numbers of T cells.
- CX5461 in clinical trials could be included in the discussion
- Typo: Results section, first paragraph, second sentence "morphologywas"

Referee #2 (Comments on Novelty/Model System for Author):

Clarity: although I had some exposure to rDNA regulation during my earlier scientific life, it took me awhile to get into the topic and I was forced to do some background reading. For the NFAT part I could easily see that others, not so familiar, would also struggle.

Interest for the nonspecialist: Surely, it is remarkable that the well-studied transcription factor NFAT has its function in the rRNA transcription and that the less studied NF45/NF90 takes part in this process. For others, this might be details.

However, elucidating this surely covers the need of developing better T-cell inhibitors for transplantation and maybe even for autoimmune-diseased patients. Therefore, the authors should be encouraged to overwork the manuscript. Then it will be suited for this journal.

Referee #2 (Remarks for Author):

EMM-2020-12834

New immune suppressants with less side effects are needed and the documented application of Pol I inhibition by CX5461 looks promising in animal models for skin and heart transplantation. Further, it appears logic that when T cells are highly activated and responsible for transplant rejection that their inhibition through an otherwise ubiquitous regulatory mechanism is decisive. Here it is to appreciate that the authors want to understand the molecular mechanism relevant in T cells. Only with this, adjusted therapies can be developed.

For this, the authors of "NF45/NF90-mediated rDNA transcription provides a novel target for immunosuppressant development" focus on NF45/NF90, a nuclear heterodimeric complex, originally found in Jurkat cells and described to bind to an NFAT response element in the IL-2 promoter. Former studies by others also demonstrated that NF90-deficient T cells cannot produce IL-2 anymore, implying a direct role of NF45/NF90 on IL2 transactivation. Interestingly, the authors of the present manuscript reveal that the rDNA promoters harbor similar 'ARRE/NFAT response elements' and accordingly bind NF45/NF90. And now, Pol I inhibition, i.e. suppression of rDNA transcription, led to similar effects as calcineurin and thus NFAT inhibition by FK506. In theory, FK506 would inhibit NFAT-mediated IL-2 expression, while CX5461 blocks Pol1-dependent rRNA synthesis. However, the authors also claim an involvement of NFAT in the nucleolus as they found it bound to the rDNA promoter. This is an interesting twist.

To my opinion, however, before this can be published, several points should be addressed. I will just go along the manuscript and mention major and minor points. In general, though, it did not become completely clear, how the heterodimeric complex NF45/NF90 regulates T-cell activation together with NFATc2 and Pol I in nucleoli and how much this contributes in comparison to the nuclear events. Where does NF45/NF90 really bind, in case the promoter-luciferase construct did not harbor the ARREs? Are the ARREs decisive and when and how does NFATc2 bind to those? Is it one complex at the same DNA element or cooperative binding to a composite site. Is the binding of the different components subsequent, together or competitive?

Introduction:

- Some of the citations could be exchanged by better / more general fitting review articles. Already the first one (Skeens et al, 2019) is cited to document the need of T-cell activation under many circumstances, but it is a particular study on pediatric patients developing GvHD.
- Second paragraph introducing NFAT: T cells express three out of four Ca-regulated family members, NFATc1, NFATc2 and NFATc3, and not just c1 and c2. By the way, this is the official nomenclature, you used otherwise in the manuscript already. So put NFAT1... in brackets here.
- It should be ARRE2 (two Rs), when introducing the abbreviation

-

M&M:

- Please provide some more information on the pHrD-IRES-Luc. The search with the citation made it necessary to go backwards to even earlier citations. Here, it seems essential to give more information about the promoter and if the ARRE sites are present - probably not - and how it relates to the sequence given in Fig. 2C.

- Please make up your mind, if you want to talk about siRNAs or shRNAs, should be the same throughout the manuscript

- Nucleolus Fractionation: Citation is not sufficient; please include a sentence "In brief..." What seems important is, whether you recovered the nucleoli and nucleoplasm (excluding nucleoli as mentioned in the cited method) or if you only gained nucleoli and compared to whole nuclear extracts as you said and labeled the figures.

- Flow... how is the anti-CD3 antibody used? Is that a pre-gate? Please provide gating strategies starting with SSC/FSC, single cells... with primary flow cytometry plots.

- Isolation of PBMCs and CD3+ T: why do you pre-culture the PBMCs in IL-2-containing media before the isolation of T cells and for how long? The sentence "The untargeted CD3+T cells were poured and collected the liquid" is not easy to understand. Please rephrase.

- The oligo sequences for EMSA are at least misleading, if not wrong. They do not contain the upper ARRE site and cannot hybridize to each other. In addition, they do not match with the grey underlined part in Fig. 2C, which you claimed to be the oligo in the legend. Maybe, I do not understand, but make it easier for the readership. Additionally, please give detailed information about the mutant oligo and also the mutant promoter luciferase construct.
- Primer table: please use the gene names instead of the proteins (for example, β -actin  Actb). Please provide primer pairs for Irfng and NF- κ B. The latter needs specification on what family member of the NF- κ B family you detected (c-Rel, p65...?)
- Luciferase: what is a 'passive' lysis buffer?

Results

- Fig. 1A: NF45/NF90 looks predominantly present in nucleoli, whereas in Fig. S1A the nuclear fraction contains high amounts of NF45/NF90 just like Fibrillarin. As said for M&M, is this a nuclear extract or the nucleoplasm devoid of nucleoli? This is important for understanding to what extent I12 or other nuclear genes could be regulated by NF45/NF90 in comparison to rDNA/RNA regulation. Therefore, if 'nuclear' is the right term, please include nucleoplasm as an extract for comparison.
- Fig. 1B and C. Controls for knock-down and overexpression would be nice.
- Fig. 1D. According to you, arrows are supposed to indicate the primer for qRT-PCR, but 'primer' in the graph labels a line. What is the arrow pointing into the 18S? What are the lines above 18S, 5.8S and 28S resembling? 47S is less known than 45S rRNA, at least 41S is derived from 45S and only indirectly from 47S rRNA. Please, redraw.
- "... truncated the NLS motif to prevent its location in the nucleolus": do you mean "... into the nucleus" or how does this motif regulate nucleolar over nuclear localization?
- In Fig. S2C the NLS mutant is clearly cytoplasmic, while the nucleoli are reduced in number, but enlarged and still merging with NP90 - puzzling! Please, explain.
- Fig. 2A: it is an unusual depiction with the promoter downstream. Even if the rDNA comes in repeats, the individual promoter should be upstream of the transcriptional start site. Alternatively, is the ARRE-containing element an enhancer and not part of the actual promoter? Please order correctly and indicate 'core promoter', 'upstream control element' and the ARRE-containing part. Fig. 2C is too fuzzy and hard to read clearly.
- Fig. 2DE: how do the mutated oligo and promoter look? (Already asked for M&Ms.)
- Fig. 3D: it seems that less protein results also in less phosphorylated protein, not so much that phosphorylation is reduced on its own.
- Fig. S3A/B: what kind of ubiquitination is that? Usually, it should be a smear caused by the Ub-chains of different length, especially when you link their ubiquitination to degradation. Then it would be a K48 ubiquitination with long chains. Only siNF90#1 might do the trick. Regarding the loss of Pol I in the extracts, do you suggest that the half-life of the polymerase is dependent on an intact complex with UBF1 and somehow NF45/NF90?
- Treatment with CX5461, the Pol I inhibitor, caused the translocation of NF45/NF90 to the nucleoplasm? Could it be that the heterodimeric complex not only binds to ARREs, but also or even rather to the transcribed rRNA? Pol I inhibition looks like the RSD mutant in S2D.
- 'The calcineurin-NFAT stimulators PMA/ionomycin': ionomycin transports Ca across membranes and therefore indeed activates calcineurin and NFAT nuclear translocation. PMA on the other hand is a diacylglycerol analogon and activates especially PKC and subsequently NF- κ B. Thus, NFAT could be involved, but it does not have to, when cells are treated with P/I. If you want to distinguish, you have to use PMA, ionomycin and P/I in comparison.
- Fig. S4 is not very informative without any key molecules indicated.
- Fig. 4E: the amount of co-precipitated NFATc2 is very low. The confocals suggest a high level of NFATc2 present. Please provide an immunoblot of the lysate for NFATc2. Please include molecular weight markers to indicate specificity of the band. By the way, why did you choose NFATc2? Is it

more prominent in your Jurkat subline? Usually, NFATc1 is nicely visible after stimulation, although 3 h of P/I might be too short to detect all NFATc1 isoforms. Please, reason on your choice of NFATc2 and if you have any indication, if NFATc2 has a special function here or just stands for all expressed NFAT family members!

- Model in Fig. 5H: A more detailed legend is necessary. What do you mean by NFATp/NFATn in the nucleus? Most likely, you refer to NFAT/AP1, but the term 'NFAT nuclear' is not used for more than 20 years anymore. The same applies to NFATp, the ancient term for NFATc2. In this context, do you consider NF45/NF90 the alternative interaction partner for NFAT in nucleoli? Then they would bind to the same motif, while AP1 has its own. I would consider, whether NF45/NF90 mainly interacts via protein/protein interaction in activated T cells, on the one hand with NFAT, on the other with UBF. TCR signals could lead to replacement of NF45/NF90 at the ARREs and NFAT with its transactivation domain would increase rRNA transcription in conjunction with UBF, hold together by NF45/NF90. This should be tested somehow. I suggest co-transfection assays with ARRE-HrDNA-luciferase (WT and ARRE mutants) with NF45/NF90 and / or NFATc2.

- Fig. S6A: as mentioned for the M&M, please provide the gating strategy.

- Fig. S6B: What is meant by NF- κ B? Why did you measure NF- κ B? And why should the RNA be increased? The first regulation in the NF- κ B signaling pathway is inhibitor degradation and nuclear translocation of the preformed cytoplasmic proteins.

- Fig. S6C: 6 h stimulation seems too short for a decent collection of cytokines in the supernatant. Their genes have to be transcribed and the protein translated and secreted! When looking at only 30 % of CD69 expression, cell activation was still at an early time point or could not be high because of the transfection. Do we observe an artifact caused by dying cells here? The requested gating data of flow cytometric analyses will shed some light on this.

- Fig. 6B: please provide more information on the patients, especially immunological data on the rejection.

- Fig. 6G: IL-12 is not produced by T cells, rather DCs. Thus, you cannot claim to look at a sole T-cell effect.

- Fig. 6O: This is NOT an evaluation of Tregs. Please, include Foxp3 intracellular staining. This is essential, since indeed Tregs are limiting transplant rejection. Furthermore, Tregs have been reported to be less NFAT-dependent. If the claim of Tregs being less inhibited by CX5461, you have to come up with an explanation, especially since (i) IL-2 levels are diminished and (ii) how rDNA synthesis is/could be wired in Tregs in comparison to Tcon.

- Fig. S9C: Which LN are those?

Editor Comments:

1) a .docx formatted version of the manuscript text (including Figure legends and tables). Please make sure that the changes are highlighted to be clearly visible to referees and editors alike.

Response: Thank you for your suggestion. All changes clearly marked by red color in the revised manuscript.

2) separate figure files*

Response: Done. The figure files have been separated from manuscript text.

3) supplemental information as Expanded View and/or Appendix. Please carefully check the authors guidelines for formatting Expanded view and Appendix figures and tables at

<https://www.embopress.org/page/journal/17574684/authorguide#expandedview>

Response: Done. We have carefully checked the authors guidelines and completed the work. Hope to have met the requirements, if you have any questions, please contact us.

4) a letter INCLUDING the reviewers' reports and your detailed responses to their comments (as Word file). Also, and to save some time should your paper be accepted, please read below for additional information regarding some features of our research articles:

Response: Done, a letter including the reviewers' reports and our detailed responses has been prepared. And we have revised the manuscript according the features of the research articles. Please see the following response.

5) The paper explained: EMBO Molecular Medicine articles are accompanied by a summary of the articles to emphasize the major findings in the paper and their medical implications for the non-specialist reader. Please provide a draft summary of your article highlighting

Response: Done. We have provided a draft summary. Please see it in our revised manuscript.

6) For more information: There is space at the end of each article to list relevant web links for further consultation by our readers. Could you identify some relevant ones and provide such information as well? Some examples are patient associations, relevant databases, OMIM/proteins/genes links, author's websites, etc...

Response: We added a link of our group's website in manuscript text so that readers could contact us.

7) Author contributions: the contribution of every author must be detailed in a separate section (before the acknowledgments).

Response: Done. Please see it before the acknowledgments.

8) EMBO Molecular Medicine now requires a complete author checklist (<https://www.embopress.org/page/journal/17574684/authorguide>) to be submitted

with all revised manuscripts. Please use the checklist as a guideline for the sort of information we need **WITHIN** the manuscript as well as in the checklist. This is particularly important for animal reporting, antibody dilutions (missing) and exact p-values and n that should be indicated instead of a range.

Response: Done. We have finished a complete author checklist. In addition, the antibody dilution and n have been added in the revised manuscript text. The exact *p*-values are listed in Appendix Table S3.

9) Every published paper now includes a 'Synopsis' to further enhance discoverability. Synopses are displayed on the journal webpage and are freely accessible to all readers. They include a short stand first (maximum of 300 characters, including space) as well as 2-5 one sentence bullet points that summarize the paper. Please write the bullet points to summarize the key **NEW** findings. They should be designed to be complementary to the abstract - i.e. not repeat the same text. We encourage inclusion of key acronyms and quantitative information (maximum of 30 words / bullet point). Please use the passive voice. Please attach these in a separate file or send them by email, we will incorporate them accordingly.

You are also welcome to suggest a striking image or visual abstract to illustrate your article. If you do please provide a jpeg file 550 px-wide x 400-px high.

Response: Done. The Synopsis has been provided in a separate file.

10) A Conflict of Interest statement should be provided in the main text

Response: Done. Please see it after the acknowledgments.

11) Please note that we now mandate that all corresponding authors list an ORCID digital identifier. This takes <90 seconds to complete. We encourage all authors to

supply an ORCID identifier, which will be linked to their name for unambiguous name identification.

Currently, our records indicate that the ORCID for your account is 0000-0002-0954-5600.

<https://embomolmed.msubmit.net/cgi-bin/main.plex?el=A2EI4niX1A5DCIT6Bh5B9f td5bMDIM5PBxeq3lstkq26hwY>

Response: Done. The ORCIDs of all authors have been provided and linked to their names. Please see the following list.

The ORCID of author : Hsiang-i Tsai 0000-0002-4233-1428

The ORCID of author : Xiaobin Zeng 0000-0002-3111-9202

The ORCID of author: Longshan Liu 0000-0001-5199-076X

The ORCID of author: Shengchang Xin 0000-0001-5177-5699

The ORCID of author: Yingyi Wu 0000-0003-3966-785X

The ORCID of author: Zhanxue Xu 0000-0003-2791-3101

The ORCID of author: Huanxi Zhang 0000-0001-7485-4496

The ORCID of author: Gan Liu 0000-0002-6165-3799

The ORCID of author: Zirong Bi 0000-0002-9465-8035

The ORCID of author: Dandan Su 0000-0003-4549-2339

The ORCID of author: Min Yang 0000-0003-4271-113

The ORCID of author: Yijing Tao 0000-0002-3578-4007

The ORCID of author: Changxi Wang 0000-0002-1282-3341

The ORCID of author: Jing Zhao 0000-0001-5177-5699

The ORCID of author: John E. Eriksson 0000-0002-1570-7725

The ORCID of corresponding author: Wenbin Deng 0000-0002-8897-4891.

The ORCID of corresponding author: Fang Cheng 0000-0002-8260-9244.

The ORCID of corresponding author: Hongbo Chen 0000-0002-0954-5600

Comments of Referee #1

Referee #1 (Remarks for Author):

In this manuscript the authors presented a new pathway for the NF45/NF90 complex, which was initially known to mediate rDNA transcription. They demonstrated with very convincing data that NF45/NF90 activation contributes to T cell activation and could be blocked by the transcription inhibitor CX5461. They showed successful in vitro and in vivo blockade of T cell activation by CX5461 which may prevent graft-versus-host disease.

The manuscript is well written and the experimental setup is comprehensible. The authors added further knowledge to the role of the NF45/NF90 complex, which is important to the field as well as for less specialized audience. I am not sure if the authors are aware of the clinical trials with CX5461, which would further strengthen their postulated use in a clinical setting.

Minor points:

Question 1: Data analyzing T cell population (Figure 6) could be supported by absolute numbers of T cells.

Response: Thanks! We agree with this suggestion that in addition to analyzing the ratio of CD4⁺ and CD8⁺ T cells, the number of CD3⁺ T cells in the recipient mouse spleen is also important for the result analysis. Based on the suggestion, the rate of T cells in spleen mononuclear cell suspensions was further analyzed. The following Figure 1 in response letter (named as rFigure 1) shows that CD3⁺ T cells significantly increase in DMSO-treated allogeneic heart transplantation group compared to NC

group (non-surgical group), and CX-5461 treatment significantly suppressed the CD3+ T cells proliferation. The same trend was observed in skin transplantation model (As shown in Figure 6F in revised manuscript).

We have added the figure and description for the CD3+ T cells in the revised Figure 6F, 6K and Manuscript Text.

Figure 1 (Figure 6K in MS). CX5461 significantly suppressed the T cell proliferation in heart allograft model.

A. FACS analysis of spleen T cells labeled with CD3 antibodies.

B. FACS analysis of spleen T cells labeled with CD4 and CD8 antibodies.

Question 2: CX5461 in clinical trials could be included in the discussion

Response: Thank you for your suggestion. We have renewed the discussion as your suggestion.

CX5461 is an effective and selective anticancer drug, which has been shown to inhibit rRNA transcription by selectively targeting the RNA polymerase I transcription mechanism (Ismael *et al*, 2019). In a Phase I clinical trial for advanced hematologic malignancies, CX-5461 was safe at doses associated with clinical benefit and

dermatologic adverse effects are manageable. Excitingly, antitumor activity was observed in TP53 wild-type and mutant malignancies in this trial, consistent with preclinical data. (Clinical trial information: ACTRN12613001061729) (Khot *et al*, 2019). CX-5461 is currently in Phase I clinical trials for triple negative or BRCA-deficient breast cancer in order to confirm the recommended phase II dose and schedule of CX5461 in patients with solid tumours (Clinical trial identification: NCT02719977) (Hilton *et al*, 2018).

Question 3: Typo: Results section, first paragraph, second sentence "morphologywas"

Response: Done. The typo error has been corrected. Thanks for your carefully review.

Comments of Referee #2

Referee #2 (Comments on Novelty/Model System for Author):

Clarity: although I had some exposure to rDNA regulation during my earlier scientific life, it took me awhile to get into the topic and I was forced to do some background reading. For the NFAT part I could easily see that others, not so familiar, would also struggle.

Interest for the nonspecialist: Surely, it is remarkable that the well-studied transcription factor NFAT has its function in the rRNA transcription and that the less studied NF45/NF90 takes part in this process. For others, this might be details.

However, elucidating this surely covers the need of developing better T-cell inhibitors for transplantation and maybe even for autoimmune-diseased patients. Therefore, the authors should be encouraged to overwork the manuscript. Then it will be suited for this journal.

Referee #2 (Remarks for Author):

EMM-2020-12834

New immune suppressants with less side effects are needed and the documented application of Pol I inhibition by CX5461 looks promising in animal models for skin and heart transplantation. Further, it appears logic that when T cells are highly activated and responsible for transplant rejection that their inhibition through an otherwise ubiquitous regulatory mechanism is decisive. Here it is to appreciate that the authors want to understand the molecular mechanism relevant in T cells. Only with this, adjusted therapies can be developed.

For this, the authors of "NF45/NF90-mediated rDNA transcription provides a novel target for immunosuppressant development" focus on NF45/NF90, a nuclear heterodimeric complex, originally found in Jurkat cells and described to bind to an NFAT response element in the IL-2 promoter. Former studies by others also demonstrated that NF90-deficient T cells cannot produce IL-2 anymore, implying a direct role of NF45/NF90 on IL2 transactivation. Interestingly, the authors of the present manuscript reveal that the rDNA promoters harbor similar 'ARRE/NFAT response elements' and accordingly bind NF45/NF90. And now, Pol I inhibition, i.e. suppression of rDNA transcription, led to similar effects as calcineurin and thus NFAT inhibition by FK506. In theory, FK506 would inhibit NFAT-mediated IL-2 expression, while CX5461 blocks Pol1-dependent rRNA synthesis. However, the authors also claim an involvement of NFAT in the nucleolus as they found it bound to the rDNA promoter. This is an interesting twist.

To my opinion, however, before this can be published, several points should be addressed. I will just go along the manuscript and mention major and minor points. In

general, though, it did not become completely clear, how the heterodimeric complex NF45/NF90 regulates T-cell activation together with NFATc2 and Pol I in nucleoli and how much this contributes in comparison to the nuclear events. Where does NF45/NF90 really bind, in case the promoter-luciferase construct did not harbor the ARREs? Are the ARREs decisive and when and how does NFATc2 bind to those? Is it one complex at the same DNA element or cooperative binding to a composite site. Is the binding of the different components subsequent, together or competitive?

Response: Thank you very much for your very constructive comments! We have conducted further experimental studies on these problems and provided a point-to-point reply to all comments. We believe the manuscript has been greatly improved. Once again, thank you for the kind advice.

Introduction

Question 1: Some of the citations could be exchanged by better / more general fitting review articles. Already the first one (Skeens et al, 2019) is cited to document the need of T-cell activation under many circumstances, but it is a particular study on pediatric patients developing GvHD.

Response: Some citations have been updated in our revised manuscript based on your suggestion.

Question 2: Second paragraph introducing NFAT: T cells express three out of four Ca-regulated family members, NFATc1, NFATc2 and NFATc3, and not just c1 and c2. By the way, this is the official nomenclature, you used otherwise in the manuscript already. So put NFAT1... in brackets here.

Response: We greatly appreciate this professional comment. We have updated the description of NFAT family members in Introduction section. Please see the words in red of second paragraph.

Question 3: It should be ARRE2 (two Rs), when introducing the abbreviation.

Response: Thanks! We have made correction according to the comment.

Materials and Methods

Question 1: Please provide some more information on the pHrD-IRES-Luc. The search with the citation made it necessary to go backwards to even earlier citations. Here, it seems essential to give more information about the promoter and if the ARRE sites are present - probably not - and how it relates to the sequence given in Fig. 2C.

Response: First of all, thank you very much for your careful examination. We did make a mistake in the description of rDNA promoter-luciferase construct in the M&M section. We have used pHrD-IRES-Luc in our previous published paper, so we may have misquoted it when we wrote the Material and Method. In fact, the rDNA-luciferase construct used in this study a new construct, which includes about 2000 bp upstream of the rRNA transcription initiation site and a putative ARRE2-like sequences at nucleotide positions -1830~-1757 and a NFAT binding site at -795 to -799 from the transcription start site.

A detailed description for rDNA-luciferase construct used in this study is given again in the revised M&M section and Figure 2C. I believe it can give readers a clearer understanding. Thank you again for your careful review.

Question 2: Please make up your mind, if you want to talk about siRNAs or shRNAs, should be the same throughout the manuscript

Response: Done. SiRNAs have been replaced by shRNAs throughout the manuscript.

Question 3: Nucleolus Fractionation: Citation is not sufficient; please include a sentence "In brief..." What seems important is, whether you recovered the nucleoli and nucleoplasm (excluding nucleoli as mentioned in the cited method) or if you only gained nucleoli and compared to whole nuclear extracts as you said and labeled the figures.

Response: Firstly, citation and brief description about nucleolus fractionation method have been added in the revised M&M. Secondly, we repeated the experiment and separated the nucleus, nucleoplasm and nucleolus. As shown in Figure S1, NF45 and NF90 showed a predominantly localization in nucleolus, compared to nucleoplasm (lamin B is a nucleoplasm marker and fibrillarin is a nucleolus marker).

Figure 2 (Appendix Fig S1A in MS). NF45/NF90 is a nucleolar protein complex. HeLa cells were fractionated into cytoplasmic, nuclear, nucleoplasm and nucleolar fractions, and protein extracts were analyzed using western blotting. Fibrillarin is a nucleolar marker; Lamin B is a nucleoplasm marker.

In addition, confocal microscopy also showed a predominantly nucleolar localization of NF45/90. ChIP, EMSA and luciferase assays confirmed that NF45/NF90 associates with the rDNA promoter (Figures. 2B-E in MS). More importantly, based on your comment, the assays about the binding of NF45/NF90 to IL-2 promoter or rDNA promoter were further performed. The results showed that P/I stimulation did not significantly alter the binding affinity of NF45/NF90 to the IL-2 promoter (Appendix Fig S6A), but significantly enhanced the binding of NF45/NF90 with the rDNA

promoter (Appendix Fig S6A). Moreover, NF90 did not show the same potential as NFAT overexpression to increase the level of IL-2 promoter activity (Appendix Fig S6B). All data indicated that for the contribution of NF45/NF90 to the activation of T cells, the role of NF45/NF90 in the nucleolus is much greater than that in the nucleoplasm.

Question 4 and Question 5:

Isolation of PBMCs and CD3⁺ Ts: why do you pre-culture the PBMCs in IL-2-containing media before the isolation of T cells and for how long? The sentence "The untargeted CD3⁺T cells were poured and collected the liquid" is not easy to understand. Please rephrase.

Flow... how is the anti-CD3 antibody used? Is that a pre-gate? Please provide gating strategies starting with SSC/FSC, single cells... with primary flow cytometry plots.

Response: We are sorry that some of our initial statements caused confusion to the reviewers. The paragraph and content about "Isolation of PBMCs and CD3⁺ Ts, and CFSE, CCK-8, Flow Cytometric Analysis" have been reorganized and written in revised M&M section, which hope will make it easier for readers to understand.

For this confuse "why do you pre-culture the PBMCs in IL-2-containing media before the isolation of T cells and for how long?", we did not pre-culture the PBMC in the IL-2-containing media before the isolation of T cells, the CD3⁺ T cells were directly from fresh isolated PBMC using a CD3⁺ separation kit. In some experiments, for example CCK8 assay, PBMC was cultured in R10 medium. But this sentence "Cells were cultured in R10 medium (RPMI supplemented with 10% FBS, L-glutamine, penicillin/streptomycin and 10 ng/mL IL-2)" should not be placed in the description

of CD3⁺ T cells separation process, which will cause confusion to readers. In the revised manuscript, we have described it again. Hope to make readers understand the experiment process more clearly.

For this confuse "The untargeted CD3⁺T cells were poured and collected the liquid", we have rephrase in the revised manuscript. The CD3⁺ T cell isolation kit we used is MojoSort™ Human CD3⁺ T cell Isolation Kit, please see the manual (<https://www.biolegend.com/protocols/mojosort-isolation-kits-protocol-1/4599/>)

For “Flow... how is the anti-CD3 antibody used?”, anti-CD3 antibody is used to pre-gate T cells. When using flow cytometry analysis, we first circled the cells isolated from fresh PBMC using MojoSort™ kit with SSC/FSC, then labeled the T cells with CD3+ antibody, and then proceeded with subsequent experimental analysis.

For example, the gating process for detecting CD69 expression is as follows:

Figure 3 (Figure 5A in MS). NF45/NF90 knockdown significantly inhibited the early activation of T cells.

CD69 expression, an early marker of lymphocyte activation, in CD3⁺ T cells upon stimulation with P/I for 12 h.

Question 6: The oligo sequences for EMSA are at least misleading, if not wrong. They do not contain the upper ARRE site and cannot hybridize to each other. In addition, they do not match with the grey underlined part in Fig. 2C, which you claimed to be the oligo in the legend. Maybe, I do not understand, but make it easier for the readership. Additionally, please give detailed information about the mutant oligo and also the mutant promoter luciferase construct.

Response: The wild-type luciferase construct contains a 2023 bp sequence (+24~-1999) of rDNA promoter. And the mutant luciferase construct deleted two NFAT consensus sites of GGAAA at -1830~-1726 and -799~-795.

Probes for EMSA were obtained from wild or mutant promoter luciferase construct by PCR using the following primers. Forward: 5'-TGGAGACACGGGCGGCCCCCT-3'; Reverse: 5'-TTTATCGACGATCCCTTCTTTA-3'. The red letters in grey shaded area indicate the probe sequences in Figure 2C in MS.

Thanks for your suggestion, we have further described these information in the revised M&M section, hoping to make it easier for readers to understand.

Question 7. Primer table: please use the gene names instead of the proteins (for example, β -actin  Actb). Please provide primer pairs for Ifng and NF- κ B. The latter needs specification on what family member of the NF- κ B family you detected (c-Rel, p65...?) .

Response: Thanks for your suggestion, protein names have been replaced by the gene names in the revised primer table.

In order to avoid the confusion of readers, we deleted the result of NF- κ B. In addition, the levels of IL-2 and IFN- γ were measured again by ELISA in revised MS.

Question 8: Luciferase: what is a 'passive' lysis buffer?

Response: In the luciferase assays, an assay kit from Promega was used (https://worldwide.promega.com/products/luciferase-assays/reporter-assays/dual_luciferase-reporter-assay-system/?catNum=E1910, Promega, Madison, WI, USA). In this kit, the lysis buffer is called Passive buffer. Passive lysis buffer robust lytic performance is of equal benefit when harvesting adherent cells cultured in standard dishes using active lysis. In addition to its lytic properties, Passive lysis buffer is designed to provide optimum performance and stability of the firefly and Renilla luciferase reporter enzymes.

In order to avoid readers' doubts about passive buffer, we removed the word "Passive" and added the description for the kit source in the revised manuscript.

Results

Question 1: Fig. 1A: NF45/NF90 looks predominantly present in nucleoli, whereas in Fig. S1A the nuclear fraction contains high amounts of NF45/NF90 just like Fibrillarin. As said for M&M, is this a nuclear extract or the nucleoplasm devoid of nucleoli? This is important for understanding to what extent Il2 or other nuclear genes could be regulated by NF45/NF90 in comparison to rDNA/RNA regulation. Therefore, if 'nuclear' is the right term, please include nucleoplasm as an extract for comparison.

Response: Based on your suggestion, the nucleus, nucleoplasm and nucleolus fractions were prepared and subjected to western blot. As shown in the above rFigure 2 (Appendix Fig S1A), NF45 and NF90 showed a predominantly localization in nucleolus, compared to nucleoplasm (Lamin B1 is a nucleoplasm marker and fibrillarin is a nucleolus marker).

We agree with you that this is important for understanding to what extent IL-2 or other nuclear genes could be regulated by NF45/NF90 in comparison to rDNA regulation. Western blot result is consistent with the observation of confocal microscopy (Figure 1 in MS). In addition, ChIP, EMSA and luciferase assays confirmed that NF45/NF90 binds to the rDNA promoter (Figures. 2B-E in MS). More importantly, based on your comment, the assays about the binding of NF45/90 to IL-2 promoter or rDNA promoter were further performed. The results showed that P/I stimulation did not significantly alter the binding affinity of NF45/NF90 to the IL-2 promoter (Appendix Fig S6A), but significantly enhanced the binding of NF45/NF90 with the rDNA promoter (Appendix Fig S6A). Moreover, NF90 did not show the same potential as NFAT overexpression to increase the level of IL-2 promoter activity (Appendix Fig S6B). All data indicated that for the contribution of NF45/NF90 to the activation of T cells, the role of NF45/NF90 in the nucleolus is much greater than that in the nucleoplasm.

All data indicated that for the contribution of NF45/90 to the activation of T cells, the role of NF45/NF90 in the nucleolus is much greater than that in the nucleoplasm.

Question 2: Fig. 1B and C. Controls for knock-down and overexpression would be nice.

Response: Done. Controls for knock-down and overexpression have been performed in the revised manuscript.

Question 3: Fig. 1D. According to you, arrows are supposed to indicate the primer for qRT-PCR, but 'primer' in the graph labels a line. What is the arrow pointing into the 18S? What are the lines above 18S, 5.8S and 28S resembling? 47S is less known than 45S rRNA, at least 41S is derived from 45S and only indirectly from 47S rRNA. Please, redraw.

Response: Thanks! Firstly, based on your suggestion, 47S pre-rRNA has been replaced by 45S pre-rRNA throughout the manuscript. Secondly, we have redrawn Figure 1D. In the revised figure, arrow indicates the cleavage site of 45S pre-rRNA to form 41S pre-rRNA, and the position of the primer is indicated by a short line. At the same time, the lines above 18S, 5.8S and 28S have been removed. Please see Figure 1D in the revised MS.

Question 4 and Question 5: "... truncated the NLS motif to prevent its location in the nucleolus": do you mean "... into the nucleus" or how does this motif regulate nucleolar over nuclear localization?

S2C the NLS mutant is clearly cytoplasmic, while the nucleoli are reduced in number, but enlarged and still merging with NP90 - puzzling! Please, explain.

Response: So far, we do not know which amino acid sequences will affect the nucleolar localization but not the nucleocytoplasmic localization of NF90. Therefore, this study only deleted the possible NLS in order to reduce their nuclear localization and see if they can affect the regulation of rDNA transcription.

As can be seen from Appendix Figure S2C, the deletion of NLS significantly affected the nuclear localization of NF90, making most of NF90 dispersed in the nucleoplasm. In addition, the same as you said, we also observed that although NLS significantly reduced the number of nuclear NF90, a small amount of NF90 remained in the nucleus. Moreover, to our surprise, the NF90 mutant retained the nucleus is still mainly located in the nucleolus rather than in the nucleoplasm, which indicated that in addition to NLS sequence, there are other sequences that can regulate the nucleolar localization of NF90. In any case, the remaining NF90 in the nucleus is still mainly located in the nucleolus, which further strengthens the evidence that NF90 mainly plays a role in the nucleolus rather than in the nucleoplasm.

Question 6: Fig. 2A: it is an unusual depiction with the promoter downstream. Even if the rDNA comes in repeats, the individual promoter should be upstream of the transcriptional start site. Alternatively, is the ARRE-containing element an enhancer and not part of the actual promoter? Please order correctly and indicate 'core promoter', 'upstream control element' and the ARRE-containing part. Fig. 2C is too fuzzy and hard to read clearly.

Response: Based on your suggestion, we have redrawn the model diagram, please see the following rFigure 4 (Figure 2A in MS).

rFigure 4 (Figure 2A in MS). Schematic illustration of a single human rDNA repeat and positions of the primers used for ChIP.

In addition, we also have marked the core promoter, upstream regulating sequences and the ARRE-like site in the following rFigure 5 (Figure 2C in revised MS). The red letters in grey shaded area indicate the probe region used in EMSA assay, in which has an ARRE2-like sequence. Grey shaded area indicates the upstream control element. Blue color indicates the core promoter region.

The previous report revealed that The ARRE-2 site in the IL-2 promoter is extremely well conserved between species. Comparison of the human, bovine, and murine ARRE-2 elements in the IL-2 promoter reveal three conserved regions, including the 5'purine (NFAT binding site), core (AP-1 binding site) and 3'purine sequences. Both of these elements have been demonstrated to be essential for maximal ARRE-2 transcriptional activity (Nirula *et al*, 1997). We analyzed the promoter of rDNA gene and found that there was a similar ARRE2 region (Named ARRE2-like sequence). The conserved elements including the 5' purine, core, and 3' purine sequences are boxed in the rDNA promoter, and the sequences that are critical for NFAT and AP-1 binding are indicated.

rFigure 5 (Figure 2C in revised MS). The human rDNA promoter region.

The red letters in grey shaded area indicate the probe region used in EMSA assay, in which has an ARRE2-like sequence. Grey shaded area indicates the upstream control element. Blue color indicates the core promoter region.

Question 7: Fig. 2DE: how do the mutated oligo and promoter look? (Already asked for M&Ms.)

Response: The wild-type luciferase construct contains a 2023bp sequence (+24~-1999) of rDNA promoter. And the mutant luciferase construct deleted two NFAT consensus sites of GGAAA at -1830~-1826 and -799~-795.

Probes for EMSA were obtained from wild or mutant promoter luciferase construct by PCR using the following primers. Forward: 5'-TGGAGACACGGGCCCGCCCCCT-3'; Reverse: 5'-TTTATCGACGATCCCTTCTTTA-3'. The sequence of probe matches the red letters in the grey shaded area in the above rFigure 5 (Figure 2C in revised MS).

Question 8: Fig. 3D: it seems that less protein results also in less phosphorylated protein, not so much that phosphorylation is reduced on its own.

Response: We agree with your comment. From Fig. 3D, we can see the less phosphorylated UBF protein was mainly caused by the reduced content of total UBF protein that is related to NF90 knockdown.

Question 9: Fig. S3A/B: what kind of ubiquitination is that? Usually, it should be a smear caused by the Ub-chains of different length, especially when you link their ubiquitination to degradation. Then it would be a K48 ubiquitination with long chains.

Only siNF90#1 might do the trick. Regarding the loss of Pol I in the extracts, do you suggest that the half-life of the polymerase is dependent on an intact complex with UBF1 and somehow NF45/NF90?

Response: Firstly, we repeated the ubiquitination experiment. As shown in rFigure 6 (Appendix Figure S3), NF90#1 knockdown indeed induced the markedly ubiquitination of UBF. Secondly, we further investigated which ubiquitination type of UBF was induced by NF90 knockdown using K48 and K63 specific antibodies. We found NF90 knockdown-mediated polyubiquitination of UBF was a type of K48-linked ubiquitination.

Based on the results (Figure 3D and rFigure 6), it is possible that the half-life of the polymerase I is dependent on an intact complex with UBF and somehow NF45/NF90.

rFigure 6 (Appendix Figure S3). NF90 knockdown caused the K48-linked polyubiquitination of UBF1 (Appendix Figure S3 in MS).

A. Western blot showing UBF and NF90 protein level *in* HEK293 cells infected with lentivirus containing NF90-specific shRNAs (shNF90#1 and shNF90#2). **B.** IP

analysis of UBF in HEK293 infected with lentivirus containing shNF90#1. **C.** IP analysis of UBF ubiquitination in HEK293 infected with lentivirus containing shNF90#1.

Question 10: Treatment with CX5461, the Pol I inhibitor, caused the translocation of NF45/NF90 to the nucleoplasm? Could it be that the heterodimeric complex not only binds to ARREs, but also or even rather to the transcribed rRNA? Pol I inhibition looks like the RSD mutant in S2D.

Response: Thank you to come up with this great speculation. We agree that we cannot exclude the possibility that NF45/90 binds to the transcribed pre-rRNA and plays a role in pre-rRNA processing or maturation, since NF45/90 has been reported to participate in miRNA biogenesis through the preferential binding to transcribed pri-microRNAs. However, the nucleolus is a sensor for cellular stresses, and many types of stresses induce nucleoplasmic translocation of nucleolar proteins, which have a NLS sequence. Thus it may not be surprising to see NF45/90 to translocate from nucleolus to the nucleoplasm upon stressed by CX5461. Furthermore, CX5461 is a Pol I inhibitor mainly inhibiting rRNA transcription step. Therefore, we believe that the major molecular mechanism of NF45/90 in T cell activation is to interact with ARRE2-like site on rDNA promoter, directly regulating rDNA transcription (supported by evidence from CHIP, EMSA and luciferase assays in Figure 2B-E and Appendix Figure S6).

Question 11: 'The calcineurin-NFAT stimulators PMA/ionomycin': ionomycin transports Ca across membranes and therefore indeed activates calcineurin and NFAT nuclear translocation. PMA on the other hand is a diacylglycerol analogon and activates especially PKC and subsequently NF- κ B. Thus, NFAT could be involved,

but it does not have to, when cells are treated with P/I. If you want to distinguish, you have to use PMA, ionomycin and P/I in comparison.

Response: Thanks for your professional comments. Based on your suggestion, Jurkat T cells were treated with DMSO, PMA, ionomycin or PMA/ionomycin (P/I) for 6 h respectively. NFATc2 subcellular distribution was then observed by confocal microscopy. Expectedly, ionomycin stimulation, but not DMSO and PMA, resulted in the significant translocation of NFATc2 from cytoplasm to nucleus and nucleolus. Moreover, it showed the more obvious nuclear and nucleolar in P/I stimulation than only ionomycin (rFigure 7, also Appendix Figure 5A).

rFigure 7 (Appendix Figure S5A). NFATc2 translocated from cytoplasm to the nucleus and nucleolus upon ionomycin or P/I stimulation.

Question 12: Fig. S4 is not very informative without any key molecules indicated.

Response: We agree with the reviewer's comment. Based on the suggestion, the data has been removed from revised MS.

Question 13: Fig. 4E: the amount of co-precipitated NFATc2 is very low. The confocals suggest a high level of NFATc2 present. Please provide an immunoblot of the lysate for NFATc2. Please include molecular weight markers to indicate specificity of the band. By the way, why did you choose NFATc2? Is it more prominent in your Jurkat subline? Usually, NFATc1 is nicely visible after stimulation, although 3 h of P/I might be too short to detect all NFATc1 isoforms. Please, reason on your choice of NFATc2 and if you have any indication, if NFATc2 has a special function here or just stands for all expressed NFAT family members!

Response: Initially, NFATc2 was just selected to stands for all expressed NFAT family members. Based on your suggestion, we also investigated the response of another key molecular member NFATc1 upon the stimulations with PMA, ionomycin or PMA/ionomycin (P/I) for 6 h, respectively. Similar to NFATc2, ionomycin stimulation, but not DMSO and PMA, resulted in the significant translocation of NFATc1 from cytoplasm to nucleus and nucleolus. Moreover, it showed the more obvious nuclear and nucleolar in P/I stimulation than only ionomycin (**rFigure 8**).

rFigure 8 (Appendix Figure S5B). NFATc1 translocated from cytoplasm to the nucleus and nucleolus upon ionomycin or P/I stimulation.

In addition, co-IP was performed to test the interaction between NF45/90 and NFAT1 or NFAT2 upon P/I stimulation. As shown in the following figures, both NF45 and NF90 can interact with NFATc1 or NFATc2 in activated T cells (rFigure 9, Figure 4F in revised MS).

Therefore, NFATc2 was chosen as a representative in our follow-up study.

rFigure 9 (Fig. 4F in revised MS). Co-IP analysis of NF45/NF90 with NFATc2 or NFATc1 in Jurkat cells upon P/I stimulation.

Question 14: Model in Fig. 5H: A more detailed legend is necessary. What do you mean by NFATp/NFATn in the nucleus? Most likely, you refer to NFAT/AP1, but the term 'NFAT nuclear' is not used for more than 20 years anymore. The same applies to NFATp, the ancient term for NFATc2. In this context, do you consider NF45/NF90 the alternative interaction partner for NFAT in nucleoli? Then they would bind to the same motif, while AP1 has its own. I would consider, whether NF45/NF90 mainly interacts via protein/protein interaction in activated T cells, on the one hand with NFAT, on the other with UBF. TCR signals could lead to replacement of NF45/NF90 at the ARREs and NFAT with its transactivation domain would increase rRNA transcription in conjunction with UBF, hold together by NF45/NF90. This should be

tested somehow. I suggest co-transfection assays with ARRE-HrDNA-luciferase (WT and ARRE mutants) with NF45/NF90 and / or NFATc2.

Response: Firstly, as the reviewer said, I really want to use NFATp/NFATn to represent the NFAT/NF45 and 90 relationship, like NFAT/AP1 in the original manuscript. However, according to the professional advice, we have given up this statement of NFATp/NFATn in the revised manuscript. Secondly, we agree with the reviewer's opinion that NF45/NF90 mainly interacts via protein/protein interaction in activated T cells, on the one hand with NFAT, on the other with UBF.

In order to confirm the relationship between NF90 and NFAT, based on your suggestion, we first co-transfected NF90 and NFATc2 and found that NF90 and NFATc2 collaboratively promote rDNA transcription after P/I stimulation. This synergistic effect was inhibited when the ARRE2-like binding site was deleted at the rDNA promoter (rFigure 10, Figure 4H in the revised MS). In addition, we found that NFATc2 cannot bind to the rDNA promoter and increase rDNA transcription in the absence of NF90 (Figure 4I). Therefore, we speculate that NF90 recruits NFATc2 into the nucleolus to regulate rDNA transcription during T cell activation. We speculate that in resting T cells, NF45/90 can bind to the ARRE2-like site of rDNA and maintain the basic level of rDNA transcription. When T cells are activated, TCR signals could lead to the translocation of NFAT to nucleoli, and NF45/NF90 is the alternative interaction partner for NFAT in nucleoli (Please see the Co-IP in the rFigure 9). The complex formation of NFAT-NF45/NF90-UBF increases rRNA transcription for resting level to activated level.

Although both NFATc2 and NF90 are able to combine ARRE2-like sites of rDNA, we proposed that the exact sequence by NFATc2 is different from NF90-binding

sequence. Deleting GGAAA, the key NFAT binding sequence in rDNA promoter, interrupt the association of NF90 and its ARRE2-like site. Possibly the exact binding sites of NF90 and NFAT on ARRE2-like sequences are physically close to each other at the rDNA promoter, thus deleting GGAAA-binding sequence would influence the binding efficiency of NF90 and corresponding ARRE2-like area.

rFigure 10 (Fig. 4H in revised MS). NFATc2 and NF90 collaboratively promote rDNA transcription after P/I stimulation.

Thirdly, the model in Fig. 5H (have removed, and replaced by new Fig 7 and Synopsis graphic) have been further improved and hope to enable readers to get more accurate information and easier to understand (please see Fig 7 or Synopsis graphic). Thanks for the reviewer's valuable comments.

Question 15: S6A: as mentioned for the M&M, please provide the gating strategy.

Response: Done. Please see the above rFigure 3. We first circled the cells isolated from fresh PBMC using MojoSort™ kit with SSC/FSC, then labeled the T cells with CD3+ antibody, and then proceeded with CD69⁺ analysis.

Question 16: Fig. S6B: What is meant by NF-kB? Why did you measure NF-kB? And why should the RNA be increased? The first regulation in the NF-kB signaling

pathway is inhibitor degradation and nuclear translocation of the preformed cytoplasmic proteins.

Response: As previous study reported NF- κ B can be activated during organ transplantation and autoimmune diseases, and also an increased mRNA level was observed (Molinero & Alegre, 2012, Uttra *et al*, 2018). So in this study, we evaluated NF- κ B. But based on your professional suggestion, in order to avoid the confusion of readers, we deleted the result of NF- κ B in revised MS.

Question 17: Fig. S6C: 6 h stimulation seems too short for a decent collection of cytokines in the supernatant. Their genes have to be transcribed and the protein translated and secreted! When looking at only 30 % of CD69 expression, cell activation was still at an early time point or could not be high because of the transfection. Do we observe an artifact caused by dying cells here? The requested gating data of flow cytometric analyses will shed some light on this.

Response: Thanks for your valuable comments.

Firstly, in a preliminary experiment, we found long term P/I stimulation, such as more than 24 hours, can cause T cell death, thus 6 hours of P/I stimulation was selected in the original study. For the CD69 expression assay, we collected the cells and performed flow cytometry analysis immediately after six hours of stimulation. As you said, only 30 % of CD69 expression, cell activation was still at an early time point. For cytokines assay, we replaced P/I-contained medium with P/I-free fresh medium after 6h stimulation and the supernatant was harvested for cytokine assay after another 24h culture. So it is not 6h but 24h for cytokine assay.

Secondly, based on the suggestion, we also performed a longer time of 12 h P/I stimulation. As shown in the following rFigure 11A, about 80 % of CD69 expression was observed in shNC group, showing a higher percentage of T cells were activated at the time point of 12h stimulation, and NF45 and NF90 knockdown significantly suppressed the expression of CD69.

We also detected cytokines level after another 24h culture. The present results are consistent with the previous ones (please see the following rFigure 11B and 11C).

rFigure 11 (Figure 5 in MS). NF45/NF90 knockdown suppressed the early activation of T cells.

A. CD69 expression, an early marker of lymphocyte activation, in CD3⁺ T cells upon stimulation with P/I for 12 h. **B-C.** IL-2 and IFN- γ levels in culture supernatants of CD3⁺ T cells with the indicated treatments at 24h after 12h of P/I stimulation were detected by ELISA assays.

Question 18: Fig. 6B: please provide more information on the patients, especially immunological data on the rejection.

Response: The patient's information has been placed in the supplementary material

S-Table 1. And a more detailed description has been added in the M&M section in the revised MS.

S-Table 1

Variables	Total (n = 23)	ABMR (n = 8)	TCMR (n = 8)	stable (n = 7)	P
Donors					
Types, n (%)					0.005
DD	13 (57)	2 (25)	8 (100)	3 (43)	
LD	10 (43)	6 (75)	0 (0)	4 (57)	
Recipients					
Gender, n (%)					0.017
Female	9 (39)	1 (12)	2 (25)	6 (86)	
Male	14 (61)	7 (88)	6 (75)	1 (14)	
Age (year), Median (IQR)	33.81 (31.40, 39.65)	37.84 (34.13, 48.32)	32.97 (30.31, 41.70)	31.72 (30.76, 33.50)	0.102
Weight (kg), Median (IQR)	52.0 (43.8, 63.8)	63.3 (52.0, 64.0)	54.3 (44.9, 63.3)	42.5 (41.4, 50.0)	0.138
Previous transplantation, n (%)					0.494
0	20 (87)	8 (100)	6 (75)	6 (86)	
1	3 (13)	0 (0)	2 (25)	1 (14)	

Days from transplant to sample, Median (IQR)	366 (140, 582)	1474 (633, 2353)	128 (103, 149)	376 (369, 378)	0.001
CNIs *, n (%)					1.000
CsA	1 (4)	0 (0)	1 (12)	0 (0)	
FK506	22 (96)	8 (100)	7 (88)	7 (100)	
FK506 C ₀ (ng/ml), Median (IQR)	6.6 (5.3, 7.8)	7.7 (7.2, 8.5)	5.7 (5.1, 6.6)	6.3 (5.6, 8.3)	0.161
CsA C ₀ (ug/L) [#]	—	—	127.5	—	—
White blood cell count (10 ⁹ /L), Median (IQR)	6.99 (5.98, 9.21)	7.00 (6.22, 8.97)	7.44 (6.39, 9.86)	6.41 (5.98, 8.25)	0.694
Lymphocyte count (10 ⁹ /L), Mean ± SD	1.41 ± 0.68	1.13 ± 0.37	1.53 ± 0.82	1.61 ± 0.75	0.339
Neutrophil (10 ⁹ /L), Median (IQR)	4.65 (3.74, 6.85)	5.05 (4.30, 7.62)	6.06 (4.29, 6.87)	3.80 (3.54, 5.65)	0.408
Monocyte count (10 ⁹ /L), Mean ± SD	0.61 ± 0.21	0.62 ± 0.18	0.68 ± 0.29	0.52 ± 0.14	0.363

CNIs, Calcineurin inhibitors; FK506, Tacrolimus; CsA, Cyclosporin A; LD, Living donor; DD, Death donor. * Standard maintenance immunosuppression, including CNI, prednisolone and mycophenolic acid was applied to all patients. [#] Only two patients received cyclosporine A, and no standard deviation of CsA C₀ was provided.

Question 19: Fig. 6G: IL-12 is not produced by T cells, rather DCs. Thus, you cannot claim to look at a sole T-cell effect.

Response: We appreciate the reviewer's helpful comments. We only want to see the overall transplant rejection phenomenon by detecting IL-12 levels. Based on your suggestion, we deleted the results of IL-12 and added the detecting results of IL-2 and IFN γ in the revised manuscript again. As shown in Figure 6G, 6H, 6L and 6M in the revised MS, CX5461 can effectively inhibit the secretion expression of IL-2 and IFN γ .

Question 20: Fig. 6O: This is NOT an evaluation of Tregs. Please, include Foxp3 intracellular staining. This is essential, since indeed Tregs are limiting transplant rejection. Furthermore, Tregs have been reported to be less NFAT-dependent. If the claim of Tregs being less inhibited by CX5461, you have to come up with an explanation, especially since (i) IL-2 levels are diminished and (ii) how rDNA synthesis is/could be wired in Tregs in comparison to Tcon.

Response: Thank you for your professional comments! Based on your suggestion, CD4⁺ CD25⁺ Foxp3⁺ Tregs in lymph nodes of recipient mice were further detected using flow cytometer. As shown in the following rFigure 12, consistent with previous reports that FK506 did not increase the proportion of Treg cells, even caused inhibition, while CX5461 treatment did resulted in a slight but statistically significant increase of Treg.

rFigure 12 (Figure 6O in MS). CX5461 treatment increased the proportion of Treg cells.

As the reviewer said, CNIs (CsA or FK506) block the TCR-induced translocation of NFAT into the nucleus, thereby blocking Teff function and IL-2 transcription (Hermann-Kleiter & Baier, 2010). Therefore, these agents have been shown to limit the availability of IL-2 as a growth factor for Tregs, resulting in a decrease in Treg numbers, as shown in liver and kidney transplant patients (Whitehouse *et al*, 2017). This phenomenon is also observed in our study that Treg cells were inhibited in FK506-treated recipient mice (**rFigure 12 in this response letter or Figure 6O in MS**).

But the regulation of Treg might be very complicated. In recent years, more and more studies indicated besides NFAT and IL-2, some important cytokines and transcription regulators, such as IL-4, TGF- β , IL-10, Blimp-1 and IRF4 also play important roles in the regulation of Treg. Another clinically used immunosuppressive drug Rapamycin (Rapa) has also been shown to promote Treg cell generation *in vivo* and was used to expand Treg cells together with low-dose IL-2 *in vitro* (Battaglia *et al*, 2006) Thus, for CX5461 we speculated that the low level of IL-2 can provides a basal survival of Tregs and CX5461 promotes the Tregs expansion *in vivo* through somehow mechanism like Rapamycin does.

Question 21: Fig. S9C: Which LN are those?

A36. These LNs are superficial cervical lymph nodes (on the surgical side). We have added a description in the revised MS.

Reference

Battaglia M, Stabilini A, Migliavacca B, Horejs-Hoeck J, Kaupper T, Roncarolo M-GJTJoI (2006) Rapamycin promotes expansion of functional CD4+ CD25+ FOXP3+ regulatory T cells of both healthy subjects and type 1 diabetic patients. **177**: 8338-8347

Hermann-Kleiter N, Baier GJB, The Journal of the American Society of Hematology (2010) NFAT pulls the strings during CD4+ T helper cell effector functions. **115**: 2989-2997

Hilton J, Cescon D, Bedard P, Ritter H, Tu D, Soong J, Gelmon K, Aparicio S, Seymour LJAoO (2018) 44O CCTG IND. 231: A phase 1 trial evaluating CX-5461 in patients with advanced solid tumors. **29**: mdy048. 003

Ismael M, Webb R, Ajaz M, Kirkby KJ, Coley HMJC (2019) The targeting of rna polymerase I transcription using CX-5461 in combination with radiation enhances tumour cell killing effects in human solid cancers. **11**: 1429

Khot A, Brajanovski N, Cameron DP, Hein N, Maclachlan KH, Sanij E, Lim J, Soong J, Link E, Blombery PJCd (2019) First-in-human RNA polymerase I transcription inhibitor CX-5461 in patients with advanced hematologic cancers: Results of a phase I dose-escalation study. **9**: 1036-1049

Molinero LL, Alegre M-LJTR (2012) Role of T cell–nuclear factor κ B in transplantation. **26**: 189-200

Nirula A, Moore DJ, Gaynor RBJJJoBC (1997) Constitutive binding of the transcription factor interleukin-2 (IL-2) enhancer binding factor to the IL-2 promoter. **272**: 7736-7745

Uttra AM, Shahzad M, Shabbir A, Jahan SJJoe (2018) Ephedra gerardiana aqueous ethanolic extract and fractions attenuate Freund Complete Adjuvant induced arthritis

in Sprague Dawley rats by downregulating PGE2, COX2, IL-1 β , IL-6, TNF- α , NF-kB and upregulating IL-4 and IL-10. **224**: 482-496

Whitehouse G, Gray E, Mastoridis S, Merritt E, Kodela E, Yang JH, Danger R, Mairal M, Christakoudi S, Lozano JJPotnaos (2017) IL-2 therapy restores regulatory T-cell dysfunction induced by calcineurin inhibitors. **114**: 7083-7088

11th Dec 2020

Dear Dr Chen,

Thank you for the submission of your revised manuscript to EMBO Molecular Medicine. We have now received the enclosed report from the referee who was asked to re-assess it. As you will see the referee is now supportive and I am pleased to inform you that we will be able to accept your manuscript pending the following amendments:

1. In the main manuscript file, please do the following:

- remove the red color font
- remove the "Supporting information" section.

2. Appendix: please remove the red color font.

3. Figure 2: if you have a better image for Figure 2D, we would recommend you to replace the current one.

4. Data availability: since this study does not generate large-scale datasets, please only include the following sentence in this section- "This study includes no data deposited in external repositories".

5. We now encourage the publication of source data, particularly for electrophoretic gels, blots, but also microscopy images with the aim of making primary data more accessible and transparent to the reader. Would you be willing to provide a PDF file per figure that contains the original, uncropped and unprocessed scans of all or key gels used in the figure (including molecular weight markers)? The PDF files should be labeled with the appropriate figure/panel number (1 file/figure), and should have molecular weight markers; further annotation may be useful but is not essential. The PDF files will be published online with the article as supplementary "Source Data" files. More information can be found here: <https://www.embopress.org/page/journal/17574684/authorguide#sourcedata>

6. As part of the EMBO Publications transparent editorial process initiative (see our Editorial at <http://embomolmed.embopress.org/content/2/9/329>), EMBO Molecular Medicine will publish online a Review Process File (RPF) to accompany accepted manuscripts.

In the event of acceptance, this file will be published in conjunction with your paper and will include the anonymous referee reports, your point-by-point response and all pertinent correspondence relating to the manuscript. Let us know if you do NOT agree with this.

7. Our data editor has made a couple of comments on your manuscript (see attached). Please address these issues and keep the track mode on.

8. I have slightly modified the synopsis text, please let me know if it is fine like this or if you would like to introduce further modifications:

This study reveals NFAT-NF45/NF90-mediated rDNA transcription as a key regulating axis in modulating T cell activation. Targeting ribosome biogenesis could be a novel immunosuppressive therapy for organ transplantation.

NF45/NF90 protein complex positively regulates rDNA transcription by directly binding to rDNA gene promoter.

Upon T cell activation, NAFT translocates to nucleolus and interacts with NF45/NF90 to cooperatively promote rDNA transcription .

Knockdown of NF45/NF90 or inhibiting rDNA transcription using a polymerase I inhibitor CX5461 suppresses T cell activation.

CX5461 treatment prolongs the survival of mouse skin or heart allografts even more than the most commonly used suppressant FK506.

I look forward to reading a new revised version of your manuscript as soon as possible.

Sincerely,
Jingyi

Jingyi Hou
Editor
EMBO Molecular Medicine

*** Instructions to submit your revised manuscript ***

To submit your manuscript, please follow this link:

Link Not Available

- 1) a .docx formatted version of the manuscript text (including Figure legends and tables)
- 2) Separate figure files*
- 3) supplemental information as Expanded View and/or Appendix. Please carefully check the authors guidelines for formatting Expanded view and Appendix figures and tables at <https://www.embopress.org/page/journal/17574684/authorguide#expandedview>
- 4) a letter INCLUDING the reviewer's reports and your detailed responses to their comments (as

Word
file).

5) The paper explained: EMBO Molecular Medicine articles are accompanied by a summary of the articles to emphasize the major findings in the paper and their medical implications for the non-specialist reader. Please provide a draft summary of your article highlighting

6) For more information: There is space at the end of each article to list relevant web links for further consultation by our readers. Could you identify some relevant ones and provide such information as well? Some examples are patient associations, relevant databases, OMIM/proteins/genes links, author's websites, etc...

7) Author contributions: the contribution of every author must be detailed in a separate section.

8) EMBO Molecular Medicine now requires a complete author checklist (<https://www.embopress.org/page/journal/17574684/authorguide>) to be submitted with all revised manuscripts. Please use the checklist as guideline for the sort of information we need WITHIN the manuscript. The checklist should only be filled with page numbers where the information can be found. This is particularly important for animal reporting, antibody dilutions (missing) and exact values and n that should be indicated instead of a range.

9) Every published paper now includes a 'Synopsis' to further enhance discoverability. Synopses are displayed on the journal webpage and are freely accessible to all readers. They include a short stand first (maximum of 300 characters, including space) as well as 2-5 one sentence bullet points that summarise the paper. Please write the bullet points to summarise the key NEW findings. They should be designed to be complementary to the abstract - i.e. not repeat the same text. We encourage inclusion of key acronyms and quantitative information (maximum of 30 words / bullet point). Please use the passive voice. Please attach these in a separate file or send them by email, we will incorporate them accordingly.

You are also welcome to suggest a striking image or visual abstract to illustrate your article. If you do please provide a jpeg file 550 px-wide x 400-px high.

10) A Conflict of Interest statement should be provided in the main text

11) Please note that we now mandate that all corresponding authors list an ORCID digital identifier. This takes <90 seconds to complete. We encourage all authors to supply an ORCID identifier, which will be linked to their name for unambiguous name identification.

Currently, our records indicate that the ORCID for your account is 0000-0002-0954-5600.

Please click the link below to modify this ORCID:
Link Not Available

12) The system will prompt you to fill in your funding and payment information. This will allow Wiley

to send you a quote for the article processing charge (APC) in case of acceptance. This quote takes into account any reduction or fee waivers that you may be eligible for. Authors do not need to pay any fees before their manuscript is accepted and transferred to our publisher.

Photos 400-800 DPI

*Additional important information regarding figures and illustrations can be found at <https://bit.ly/EMBOPressFigurePreparationGuideline>

The system will prompt you to fill in your funding and payment information. This will allow Wiley to send you a quote for the article processing charge (APC) in case of acceptance. This quote takes into account any reduction or fee waivers that you may be eligible for. Authors do not need to pay any fees before their manuscript is accepted and transferred to our publisher.

***** Reviewer's comments *****

Referee #2 (Remarks for Author):

All my concerns have been addressed adequately. Congrats!

The authors performed the requested editorial changes.

4th Jan 2021

Dear Dr. Chen,

We are pleased to inform you that your manuscript is accepted for publication and is now being sent to our publisher to be included in the next available issue of EMBO Molecular Medicine.

We would like to remind you that as part of the EMBO Publications transparent editorial process initiative, EMBO Molecular Medicine will publish a Review Process File online to accompany accepted manuscripts. If you do NOT want the file to be published or would like to exclude figures, please immediately inform the editorial office via e-mail.

Please read below for additional IMPORTANT information regarding your article, its publication and the production process.

Congratulations on your interesting work,

Jingyi Hou

Jingyi Hou
Editor
EMBO Molecular Medicine

Follow us on Twitter @EmboMolMed
Sign up for eTOCs at embopress.org/alertsfeeds

***** Reviewer's comments *****

*** ** IMPORTANT INFORMATION *** **

SPEED OF PUBLICATION

The journal aims for rapid publication of papers, using the advance online publication "Early View" to expedite the process: A properly copy-edited and formatted version will be published as "Early View" after the proofs have been corrected. Please help the Editors and publisher avoid delays by providing e-mail address(es), telephone and fax numbers at which author(s) can be contacted.

Should you be planning a Press Release on your article, please get in contact with embomolmed@wiley.com as early as possible, in order to coordinate publication and release dates.

LICENSE AND PAYMENT:

All articles published in EMBO Molecular Medicine are fully open access: immediately and freely

available to read, download and share.

EMBO Molecular Medicine charges an article processing charge (APC) to cover the publication costs. You, as the corresponding author for this manuscript, should have already received a quote with the article processing fee separately. Please let us know in case this quote has not been received.

Once your article is at Wiley for editorial production you will receive an email from Wiley's Author Services system, which will ask you to log in and will present you with the publication license form for completion. Within the same system the publication fee can be paid by credit card, an invoice, pro forma invoice or purchase order can be requested.

Payment of the publication charge and the signed Open Access Agreement form must be received before the article can be published online.

PROOFS

You will receive the proofs by e-mail approximately 2 weeks after all relevant files have been sent to our Production Office. Please return them within 48 hours and if there should be any problems, please contact the production office at embopressproduction@wiley.com.

Please inform us if there is likely to be any difficulty in reaching you at the above address at that time. Failure to meet our deadlines may result in a delay of publication.

All further communications concerning your paper proofs should quote reference number EMM-2020-12834-V3 and be directed to the production office at embopressproduction@wiley.com.

Thank you,

Jingyi Hou
Editor
EMBO Molecular Medicine

Corresponding Author Name: Wenbin Deng, Fang Cheng, Hongbo Chen

Manuscript Number: EMM-2020-12834